

# Neural network processing of holographic images

John S. Schreck[1], Gabrielle Gantos[1], Matthew Hayman[2], Aaron Bansemer[3], and David John Gagne[1]

[1]National Center for Atmospheric Research (NCAR), Computational and Information Systems Lab, Boulder, CO, USA
[2]National Center for Atmospheric Research (NCAR), Earth Observing Lab, Boulder, CO, USA
[3]National Center for Atmospheric Research (NCAR), Mesoscale and Microscale Meteorology Lab, Boulder, CO, USA

**Correspondence:** schreck@ucar.edu; mhayman@ucar.edu

**Abstract.**

HOLODEC, an airborne cloud particle imager, captures holographic images of a fixed volume of cloud to characterize the types and sizes of cloud particles, such as water droplets and ice crystals. Cloud particle properties include position, diameter, and shape. In this work we evaluate the potential for processing HOLODEC data by leveraging a combination of

GPU hardware and machine learning with the eventual goal of improving HOLODEC processing speed and performance. We present a hologram processing algorithm, HolodecML, that utilizes a neural network segmentation model and computational parallelization to achieve these goals. HolodecML is trained using synthetically generated holograms based on a model of the instrument, and predicts masks around particles found within reconstructed images. From these masks, the position and size of the detected particles can be characterized in three dimensions. In order to successfully process real holograms, we find we

must apply a series of image corrupting transformations and noise to the synthetic images used in training.

In this evaluation, HolodecML had comparable position and size estimations performance to the standard processing method, but improved particle detection by nearly 20% on several thousand manually labeled HOLODEC images. However, the particle detection improvement only occurred when image corruption was performed on the simulated images during training, thereby mimicking non-ideal conditions in the actual probe. The trained model also learned to differentiate artifacts and other impurities

in the HOLODEC images from the particles, even though no such objects were present in the training data set. By contrast, the standard processing method struggled to separate particles from artifacts. HolodecML also leverages GPUs and parallel computing that enables large processing speed gains over serial and CPU-only based evaluation. Our results demonstrate that the machine-learning based framework may be a possible path to both improving and accelerating hologram processing. The novelty of the training approach, which leveraged noise as a means for parameterizing non-ideal aspects of the HOLODEC

detector, could be applied in other domains where the theoretical model is incapable of fully describing the real-world operation of the instrument and accurate truth data required for supervised learning cannot be obtained from real-world observations.

## 1 Introduction

HOLODEC is a second-generation instrument designed for cloud particle characterization and sizing (Fugal et al., 2004; Spuler and Fugal, 2011). The instrument captures in-line holograms of cloud particles by transmitting a laser pulse between probe arms

and imaging the interference of the incident laser field and that scattered by the cloud particles onto a CCD. Unlike conventional



images, holograms capture phase information about the optical field and can therefore be computationally refocused to image each particle. Thus the HOLODEC instrument is able to capture a relatively large instantaneous sample volume (15 cm$^3$) while reconstructing an image of each particle at a transverse optical resolution of 6.5 μm. The particles' positions, sizes and shapes can then be obtained from these refocused images to investigate microscale cloud processes (e.g. Glienke et al. (2020); Desai

et al. (2021)). Unlike other cloud probe techniques, holographic imaging does not rely on scattering models (such as forward scattering probes) or detailed knowledge and modeling of instrument characteristics to generate accurate measurements (as required in optical array probes). In addition, the HOLODEC instantaneous sample volume is large enough to provide a point-like capture of liquid cloud properties. In those cases, observations do not need to be accumulated over long path lengths through a cloud, where the structure may vary considerably.

While HOLODEC is capable of capturing significant information content on the characteristics of cloud particles, processing the captured holograms remains a significant challenge. Current cloud particle hologram processing methods, e.g. as in Fugal et al. (2009), reconstruct a large number of planes along the optical path, then search for focused particles in the scene. This process is computationally expensive when applied to a large number of holograms, and often involves significant human intervention for identifying valid particles.

This work aims to evaluate the potential for machine learning to improve holographic cloud particle image processing by creating a modular processing model that can be trained without human intervention and deployed in large quantities for parallel computing on a cluster or cloud computing platform. While there have been prior works devoted to machine learning solutions for holographic image processing of particles in a distributed volume, they tend to focus on instances where the depth dimension is relatively small compared to the particle sizes so that the particle diffraction patterns are relatively localized. For

example Shimobaba et al. (2019) retrieves particles of size 20-100 μm over a depth of 1-3 cm, Zhang et al. (2022) reports applying an object detector to particles with 2-4 cm depth and Shao et al. (2020) performs position estimation of 2 μm particles over a depth of 1 mm. By contrast the HOLODEC sample volume extends over 15 cm depth so particle diffraction patterns are poorly localized on the CCD. The consequence of this is that we found no way to practically extend the processing depth of field to cover the entire sample volume depth within an exclusively machine learning framework. In addition, the large raw

holograms cannot be immediately segmented into smaller sections (as done in Shao et al. (2020) and Shimobaba et al. (2019)) that are more easily ingested by a neural network model. We also found that nonideal behavior of the actual instrument (partly attributable to the considerable range of environmental conditions the instrument must operate on the aircraft) is a key factor in developing an effective solution to processing the holograms, but this subject of non-ideal instrument behavior receives little to no attention from prior work.

In order to address the needs for processing HOLODEC data, we investigated several architectures before developing a hybrid approach that leverages standard holographic processing techniques, improved hardware utilization, and machine learning to identify particles. However, developing a technique for training the machine learning algorithm for improved performance over the existing software added further challenges. While it would be theoretically possible to perform supervised learning that attempts to achieve the same performance as the current software, this approach was deemed unacceptable by the team be-

cause its performance would be limited to that already obtained, and it would inherently learn the same (at the time unknown)



existing biases. Instead, we developed an approach using simulated data, where absolute truth may be known. While holograms can be simulated with significant accuracy using standard Fourier optics methods, the challenge in training a machine learning solution that can be applied to true data lies in capturing the non-ideal behavior of the instrument, where effects such as vignetting, laser mode structure, detector noise and non-uniform response and other unaccounted for physical processes can

result in non-deterministic noise, structure and transformations in the captured image. In this work, we address this by corrupting the simulations with a series of transformations and adding random noise. However the process of properly tuning these transformations and noise sources required a second optimization process. We used a series of manually labeled images to perform hyperparameter optimization on the tuning parameters, including the noise and transformations. The manually labeled data is simplified to only require a binary "yes" or "no" to designate if there is an in-focus particle centered in the identified

image. In this way, this effort has yet to entirely move beyond manual labels, though the requirements are greatly simplified from using manual labels for training data.

In the field of computer vision, there are many available architectures for detecting and labeling objects of interest that may be present in holograms. Of particular interest are the semantic segmentation and object detection architectures. Semantic segmentation involves an input image and output image, where the model is tasked with selecting a category label for each

pixel from the input image from a fixed set of labels. Different examples of segmentation models include those designed with convolutional neural network (CNN) layers and skip connections, such as U-networks (U-net) (Ronneberger et al., 2015a; Zhou et al., 2018) and other encoder-decoder architectures (Chen et al., 2018), models which incorporate attention layers (Li et al., 2018; Fan et al., 2020), as well as those which utilize pyramid schemes for learning the global image-level features (Zhao et al., 2017; Chen et al., 2017a; Li et al., 2018). Segmentation approaches based on CNNs are also increasingly being applied

in other areas of climate and weather forecasting (Agrawal et al., 2019; Sha et al., 2020; Ravuri et al., 2021) and for processing satellite imagery (Yuan et al., 2017; Xiao et al., 2019; Xie et al., 2020).

Object detection models predict bounding boxes around a fixed number of objects that may be present in the input image. The model also assigns each box a label chosen from a fixed set. Architectures, such as the "fast" region-based convolutional neural network (Ren et al., 2015), come equipped with a region proposal network (RPN) that first draws many boxes around

potential regions of interest in the image, then only the most confident boxes are selected, labeled, and returned by the model. RPNs and other approaches including You Only Look Once (YOLO) only require only one or two passes of data through the model to generate predictions compared with earlier sliding-window object detectors that require multiple passes and aggregation (Redmon et al., 2016). There are also architectures that both label bounding boxes and fill in segmentation masks around objects of interest (He et al., 2017).

The number of available architectures continues to grow for both approaches. As such, there are many potential ways to process holograms using neural networks. For example, Zhang et al. (2022) utilized an object-detector model for predicting the 3D coordinates of relatively localized particles, while Shao et al. (2020) showed that a U-net can accomplish the same objective with similar performance. Other efforts have focused on performing classification tasks with holograms, where decision-tree approaches (Grazioli et al., 2014; Bernauer et al., 2016) and convolutional neural networks have been investigated (Zhang

et al., 2018; Xiao et al., 2019; Touloupas et al., 2020; Wu et al., 2020). For example, Touloupas et al. (2020) used a vanilla





CNN model for classifying objects detected in the holograms as either artifact, water droplet, or ice particle. Wu et al. (2020) similarly explored using CNNs for classifying ice crystals in the holograms into different categories.

As we aimed to build a machine learning framework with improved speed and accuracy, we chose to focus on using the wide range of segmentation architectures available with the segmentation-models-pytorch software package. The segmentation models were chosen over potential object detectors for several reasons, first that they are usually smaller in size by comparison, and the recent studies by Shao et al. (2020) and Zhang et al. (2022) both showed that the model types can accomplish prediction of $(x, y)$ coordinates and estimate the $z$-coordinate with good performance (although, as noted, the results in both studies utilized depth-of-field ranges that rendered the particles much more localized by comparison to typical images obtained by HOLODEC). Segmentation models also have a potential advantage for processing holograms because a predicted mask shape may represent the 2D shape of an object when viewed in a plane, while object detectors predict the coordinates of a rectangular box and not the object's shape.

Our approach, named HolodecML, uses a trained segmentation model to obtain estimates of a particle's $(x, y)$ coordinates and size as represented by the diameter $d$. The neural network is evaluated on a series of reconstructed planes at given values of $z$, from which an estimate of the particle's distance from the detector arms is obtained using a fast post-processing algorithm. In preliminary investigations we could not find a one-step object detector or segmentation model design that could predict both the position and size of the particles using the raw HOLODEC images. HolodecML was designed to strike a balance between adding computational complexity relative to a one-stage approach, to obtain a predicted $(x, y, z, d)$ for the particles in raw HOLODEC images. The added computation complexity from the wave propagation calculations is managed through parallel and GPU computation, and by finding models with optimal performance through extensive hyperparameter optimization.

The investigation is organized as follows: In Section 2 we describe the preparation of simulated holograms and the HOLODEC data sets used. In Section 2.4, the steps involved in the processing of megapixel-sized holograms is described in detail. Section 2.5 describes how models were trained and optimized using the data sets, as well as manual labeling exercises performed by the authors on a subset of the HOLODEC holograms. In Section 3 the performance of trained models is characterized on the simulated and real HOLODEC holograms, as well as compared with the performance of existing software. Section 4 discusses the strengths and weaknesses of both the machine learning and direct approaches for processing holograms, and potential future approaches for improving the approach. Finally, in Section 5 we conclude our investigation.

## 2 Methods

### 2.1 Holographic imaging

Holograms are distinct from conventional imaging because the captured image retains phase information from the electric field scattered by the object. As a result, the image can be computationally refocused along the direction of propagation in the imaging system (referred to as the $z$ axis in this work). The holographic image is proportional to the optical intensity incident



on the detector which is described by the magnitude squared of the scattered and reference electric fields (Goodman, 2005)

$$I(x,y) = |E_R(x,y,z_c) + E_S(x,y,z_c)|^2 \tag{1}$$

$$= |E_R(x,y,z_c)|^2 + |E_S(x,y,z_c)|^2 + E_S^*(x,y,z_c)E_R(x,y,z_c) + E_S(x,y,z_c)E_R^*(x,y,z_c) \tag{2}$$

where $I(x,y)$ is the intensity captured by the camera, $E_R(x,y,z_c)$ is the electric field of a known reference wave at the camera plane $z_c$, $E_S(x,y,z_c)$ is the scattered field at the camera plane and $^*$ denotes complex conjugate. The first term, $|E_R(x,y,z_c)|^2$ is the effective intensity of the incident laser or reference field. Because the laser is collimated, it is relatively unaffected by hologram reconstruction. In most cases, the square of the scattered field, $|E_S(x,y,z_c)|^2$ is small and can be neglected. The third term $E_S^*(x,y,z_c)E_R(x,y,z_c)$ captures the conjugate field of the scattered particle. Finally, the last term

$E_S(x,y,z_c)E_R^*(x,y,z_c)$ captures the particle's scattered field modulated by the reference wave. The refocused particle image can be recovered from this last term by multiplying by the phase exponent of the reference wave.

In the case of the HOLODEC instrument, the reference wave is approximated as a plane wave, so that no corrections to the phase are applied. Also, because the HOLODEC instrument captures an inline hologram, the conjugate and true images overlap, however in reconstructing the real image, the conjugate image becomes heavily defocused and it's energy tends to be

spread out over the image plane.

Thus, in order to refocus a HOLODEC hologram at some plane $z$, we perform standard wave propagation on the hologram itself after normalizing the background. The refocused image at position $z$ is described by

$$I(x,y,z) = \mathcal{P}\left(h_c(x,y), z - z_c\right) \tag{3}$$

where $\mathcal{P}(E(x,y),z)$ is an operator that propagates an electric field $E(x,y)$ a distance $z$ and $h_c(x,y)$ is the intensity normalized

image,

$$h_c(x,y) = \frac{I(x,y)}{\langle I(x,y)\rangle} \tag{4}$$

where $\langle I(x,y)\rangle$ is the ensemble average over the previous and subsequent eight holograms and used to normalize out persistent intensity artefacts. The propagation operation is performed using a Fourier transform such that (Goodman, 2005)

$$\mathcal{P}\left(E(x,y),z\right) = \mathcal{F}^{-1}\left\{\exp\left(j\frac{2\pi z}{\lambda}\sqrt{1 - \lambda^2\rho^2}\right)\mathcal{F}[E(x,y)]\right\} \tag{5}$$

where $\lambda$ is the wavelength of the laser and $\rho$ is the radial spatial frequency coordinate, $\mathcal{F}$ is the Fourier transform operator and $\mathcal{F}^{-1}$ is the inverse Fourier transform operator. For numerical implementation, the Fourier transforms are approximated using Fast Fourier Transforms and both the spatial and frequency coordinates are discrete.

By refocusing a holographic image containing hundreds or even thousands of cloud particles, the in-focus image of each particle can be obtained. Based on the particle position, and which $z$ its in-focus image corresponds to, the particle position

can be obtained, and by processing the in-focus image, its size can also be obtained.





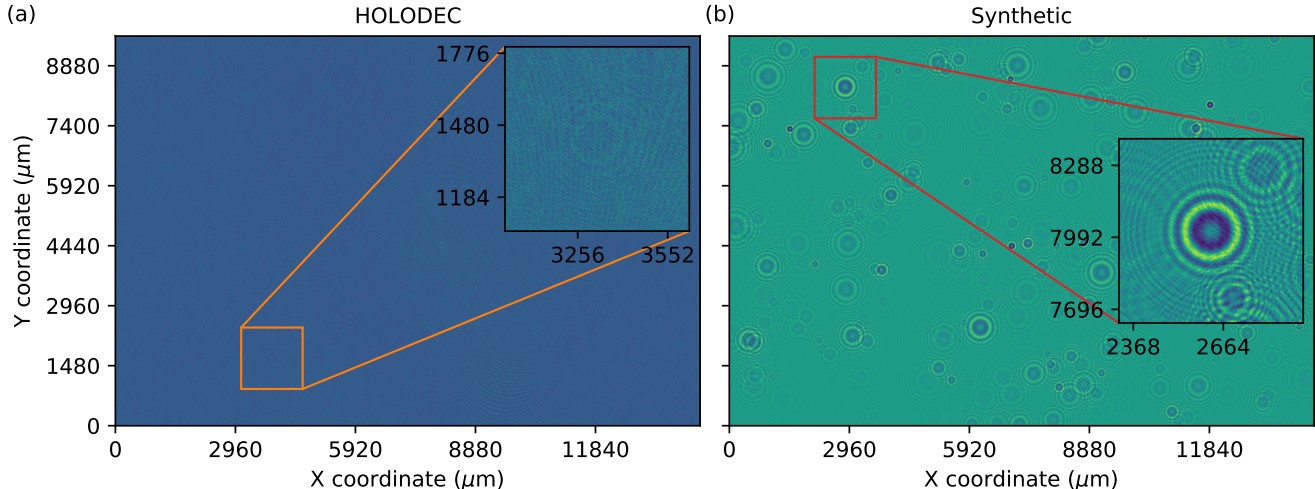

**Figure 1.** (a) An example hologram obtained from HOLODEC. (b) A synthetic hologram obtained from a simulation. The inset box in each map illustrates an out-of-focus particle.

## 2.2 HOLODEC and simulated holograms

HOLODEC operates at a wavelength of 355 nanometers. The CCD captures the holograms at in the transverse plane (e.g. the $xy$ plane) at a resolution of 2.96 micrometers per pixel in each direction. The total number of pixels in the in the $x$ and $y$ directions were 4872 and 3248 so that each hologram captures 14.42 by 9.61 millimeters, respectively. The images contain a single value for each of the pixels that may range from 0 to 255.

Figure 1(a) shows a typical hologram obtained from HOLODEC. The inset highlights an out-of-focus particle, which is positioned at its center, and shows faint rings around the particle. Close inspection of the hologram reveals other particles, as well as interference patterns. Overall the hologram appears visually faint, with the individual pixels typically varying by a small amount relative to the background.

A set of several hundred holograms were selected from the Cloud Systems Evolution in the Trades (CSET) project from June 1 to Aug 15, 2015 (Albrecht et al., 2019), in particular the RF07 subset. The selected hologram examples contained varying numbers of spherically-shaped liquid particles ranging in count from zero up to several hundred, and were largely free of ice crystals. The software that was originally used to process the RF07 data set, referred to here as "the standard method," followed the procedures described in Fugal et al. (2009), and utilized custom classification rules that were comparable to the default rules used to process other data sets in the archive (Glienke et al., 2017). The standard method was used to process the RF07 data set with a resolution along $z$ of $144\,\mu m$ and searched for particles that were positioned between 20 and 158 millimeters from the detector arms, respectively, producing a list of predicted particle positions $(x, y, z)$ and diameter $d$ for each hologram (Shaw, 2021). Two sets of ten holograms were selected from RF07 to help guide the training and validation of neural network hologram processors, and to compare the standard method against a neural network.





A set of simulated "synthetic" holograms was generated using the physical model of the instrument including the same optical settings as the holograms obtained from HOLODEC, as in Fugal et al. (2009). The synthetic holograms produced had the same size as examples in the RF07 data set. The image in Figure 1(b) shows a typical synthetic hologram. The inset similarly shows an out-of-focus particle positioned at its center, where the rings surrounding the particle are a lot easier to resolve by comparison to the HOLODEC example shown in Figure 1(a).

In total, 120 holograms were simulated that each contained 500 particles positioned at random values along $x$, $y$, and $z$, while the diameters were sampled using a gamma distribution to produce realistic particle size distributions. Along the $z$ direction, the particles were positioned between minimum and maximum values of 14.072 and 158.928 millimeters, respectively. The geometric center and the diameter of each particle were saved along with the holographic image. The simulated set was then randomly split into into training (100 holograms), validation (10 holograms), and testing (10 holograms) subsets containing in 50,000, 5,000, and 5,000 particles, respectively. The latter two subsets were used as holdout sets to test trained models as is described below.

## 2.3   Wave propagation on synthetic and HOLODEC holograms

The two particles illustrated Figure 1 are brought into focus by propagating the hologram image to some other value of $z$. Figure 2 shows the two examples at values of $z$ where each has been brought into focus. Clearly, when a particle is in focus a dark, nearly uniformly shaded circle appears from which the diameter, $d$ of the particle and its position, $(x, y, z)$, can be estimated accurately. The two examples in the figure show a relatively small and large particle in (a) and (b), respectively. When they are each out of focus, the rings that surround the particle center appear more similar in size. Reconstructed images at $z$ values just before a particle comes into focus, and just after it goes out of focus, frequently show a small bright spot (called a Poisson spot) appearing in the center of dark circle that grows in size as the image is reconstructed farther away from the particles center.

## 2.4   Processing holograms with a neural network to obtain particle positions and sizes

The processing of HOLODEC holograms consists of three main components and the overall workflow is illustrated in Figure 3. First the raw holograms are reconstructed at a set of planes along the z axis (optic axis) using standard propagation methods while leveraging GPU acceleration (Figure 3(c)(i-ii). Second, each reconstructed plane is broken into smaller images and fed into a neural network which produces a segmentation mask identifying pixels that are part of an in-focus particle (Figure 3(c)(iii-v). Finally, the smaller segmentation masks are merged (Figure 3(c)(v)) and a particle "matching" algorithm is used to identify the size and position of all detected particles in the hologram (Figure 3(c)(vii)).

Our objective was to develop a neural network processing approach that could eventually be applied to actual HOLODEC data. It was decided that there would be significant benefit in developing a processing solution that was independent of the current state-of-the-art processing software. The motivation for this is twofold. First, by creating an independent processing approach, the neural network can help identify possible biases and sources of error in the current processing package. Second, this avoided creating a solution where the standard method imposed a ceiling on the processor performance. However, in order



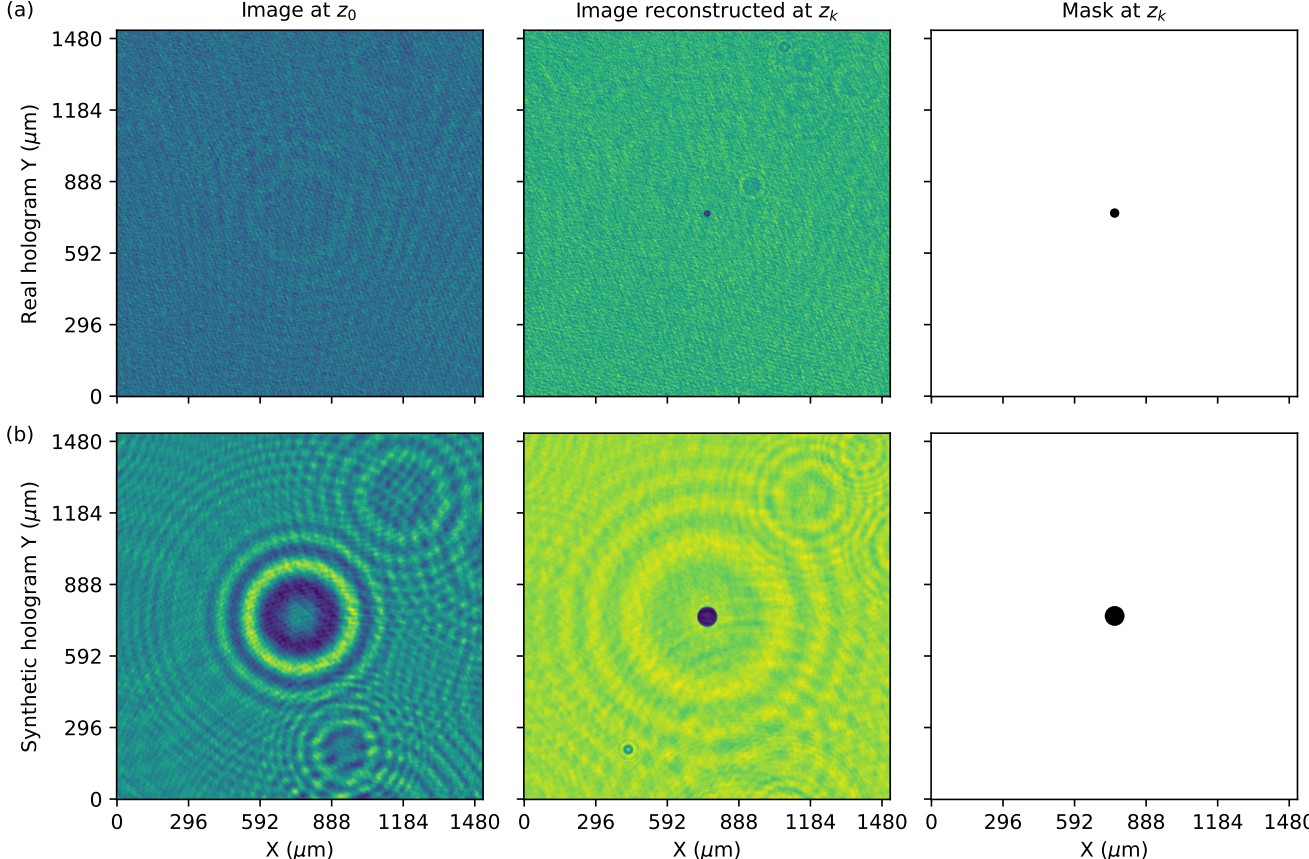

**Figure 2.** In (a) and (b) the image on the left highlights the particle from Figure 1 for the HOLODEC and synthetic examples, respectively. The panels in the center column show the same viewpoint of the particles on the left, except that each full-size image from Figure 1 has been wave-propagated to a value of $z$ where the particle is in focus in each example. In the right column the panels illustrate a "mask" with diameter equal to the particles diameter for the HOLODEC and synthetic image in (a) and (b), respectively. All examples show a dimension of 512 by 512 pixels.

to develop a processor independent of the standard method, we had to develop a training approach, illustrated in Figure 3(b), that avoided excessive manual labeling (i.e. it is unrealistic to conduct manual labeling of particle position and size over large datasets and the accuracy of such approaches would likely be suspect). In order to train the neural network, we used synthetic holograms where the true segmentation masks could be accurately and objectively defined. However, in order for the model to also work on real HOLODEC data, we also had to corrupt the synthetic holograms in a way that is reflective of real holograms (Figure 3(b)(vi)). The process of tuning the corruption of the synthetic inputs was optimized during hyperparameter optimization, which was conducted on true HOLODEC data using manually labeled image patches. Those patches were given binary labels indicating if there appeared to be an in-focus particle centered in the image (Figure 3(b)(v)). The agreement between

the processor and the manual labels was then the metric by which we established the best hyperparameters (Figure 3(b)(v,viiii-x)), while another second holdout set of manual labels was used to objectively evaluate the neural network and the standard method's performance.

### 2.4.1   Neural-network model for particle segmentation

The primary requirements for a neural network hologram processor are that it can identify a variable number particles in the hologram, and localize them so that their positions and diameters can be estimated. Figure 3(a) illustrates a neural model that can satisfy these requirements. The model takes a holographic image as input of a given size, and outputs an image the same size as the input. The output image represents the predicted probability that a pixel is contained within the boundary of an in-focus particle (right column in Figure 2). The pixel values greater than some number, typically 0.5, are labeled 1, and otherwise

are labeled 0. Thus, a mask represents a group of pixels labeled 1 that fills in the area of an in focus particle. There may be more than one mask predicted if more than one particle is in focus in the input image. From such a predicted mask, a particle's position in the plane and diameter can be estimated.

Some of the architectures available for predicting segmentation masks contain an encoder-decoder structure, as is illustrated in Figure 3(a) (labeled E and D, respectively). For example, the encoder input "head" in the widely used U-net architecture

(Ronneberger et al., 2015a; Shelhamer et al., 2017) successively down samples an input image, eventually into a latent vector of fixed size, that is then successively up-sampled back into the original image size via the decoder output head. The layers of an encoder head may also be designed to utilize layers from other pre-trained, convolutional-based image encoders, such as variants of residual neural networks (ResNet) (He et al., 2016b) trained on the ImageNet dataset, that may help speed up training and boost prediction accuracy (e.g. transfer learning). The type of segmentation model, the encoder, and whether to use

available pre-trained encoder model weights were left as hyperparameters to be optimized, and as is discussed in Section 2.5.

### 2.4.2   Grid for sub-setting megapixel-sized holograms

The full hologram sizes, when used during training and evaluation of neural models, present difficult computation, memory, and storage challenges. For example, the input size of full-size holograms is far too large to perform efficient training and inference with any neural segmentation model and a single GPU or CPU-only resources. A solution to this problem is to divide

the hologram into a grid of $N_x$ by $N_y$ square-shaped tiles. The grid can be aligned closely with the hologram dimensions if each grid tile is taken to overlap with other nearby tiles in both $x$ and $y$ directions by a fixed number pixels. The tiling procedure is illustrated schematically in Figure 3(b-c)(iii). The size of the tile and the overlap amount are referred to as the tile size and step size, respectively. We chose the tile and step sizes to be 512 and 128, respectively, hence a grid size of 38-by-25 and a total of 950 individual tiles per hologram. However, some of the tiles in either direction overshoot the hologram size with this choice.

In these cases, the tile is instead placed with one side at the boundary to prevent overshooting, and the total minimum number of needed tiles is then 828.

Each tile in the grid is passed through the neural segmentation model to obtain a predicted segmentation image containing binary labels for the pixel values, e.g. only 0s and 1s, as illustrated in Figure 3(c)(v). The model predictions are then averaged



**Figure 3.** (a) Two input examples to a neural segmentation model are shown. The segmentation model is illustrated as having encoder-decoder architecture, such as a U-net. (i) The image does not contain an in-focus particle as no mask was predicted. In (ii) the model predicted a mask around an in-focus particle. Illustrations of (b) the training and hyperparameter optimization workflow and (c) the testing / operational workflow for processing holograms.





using the appropriate grid coordinate to obtain a prediction result of the same size as the full sized holograms, illustrated in
Figure 3(c)(vi) by the reassembly step. Note that not every pixel in the predicted full-sized mask appears in the same number
of tiles due to being proximal to an edge.

### 2.4.3 Estimating $z$ through wave propagation

Neural segmentation approaches that can predict a mask around in-focus particles enables estimation the particle's center $(x, y)$
and its diameter $d$. However, the value of $z$ in the two examples shown in Figure 2 were known, and the wave propagation
operation was used to obtain the reconstructed hologram where each particle was in focus. In general, estimating the value of
coordinates and diameters for an unknown number of particles in a hologram requires evaluation of the model on $N$ recon-
structed planes at different values of $z$, as is illustrated in Figures 3(c)(ii) at the hologram propagation step. The choice of $N$
thus determines the resolution along the $z$ coordinate. Note that this is similar to how the standard method operates along $z$.

In order to obtain a resolution along $z$ to match that for $x$ and $y$ set by HOLODEC, $N$ = 48,648. As noted, the standard
method used $N$ = 1,000 to process the HOLODEC holograms which corresponds to approximately $144\,\mu m$ between recon-
structed planes. Based on the theoretically determined numerical aperture, the depth of field for the instrument is $57\,\mu m$, so we
expect limited performance improvement below this threshold. Once a choice of $N$ is made, the wave propagation calculations
involve taking the reference plane and first propagating it to $z_0$, and then to $z_1 = z_0 + \delta z$, and so on, where the values of $z_j$ are
taken to be the value at the center of the $jth$ bin along the range of $z$ values. In the results section, we compare the performance
of models trained using different values of $N$.

Note that the wave propagation calculation performed to obtain a plane at one $z$ is independent from an identical calculation
performed to obtain a plane at another $z$. Thus, if resources are available, the steps in Figure 3(c) may scale such that all
planes may be processed simultaneously. In such a scenario, the time it would take to process the entire z-range therefore
equals the time it takes to analyze one plane. Furthermore, the 828 grid tiles in a plane that get passed through a neural model
could also be processed simultaneously. We also improved the performance of the wave-propagation calculation shown in
Figure 3(b-c)(ii) by leveraging the software package PyTorch (1.9), which provides GPU support for complex numbers and
Fourier transformations. In appendix A1 we compare the neural network processing time versus the standard method.

### 2.4.4 Post-processing model predictions in 3D

Figure 3(c)(vii) illustrates the final step in the processing of holograms involves matching the reassembled hologram-sized
predictions using extracted values of $(x, y, z, d)$ for the particles identified across $N$ planes. The final result of the matching
procedure is a list of $M$ predicted particle coordinates and diameters, that can be further paired with a list of true coordinates
or those obtained from the standard method. Matching involves (1) grouping pixel slices in $N$ 2D planes that identify the
particles, and (2) computing the distance among all the identified particle slices, and putting those that fell within a specified
matching distance threshold into clusters, which is defined as the maximum distance by which two particles can be considered
members of the same cluster.





In the first step, the $(x, y)$ coordinates representing the center-of-mass of particles and the particle diameters were identified from the predicted masks. For planes containing at least one pixel labeled 1, a pixel labeled 1 was taken to be part of a mask (group) if a neighboring pixel along $x$ or $y$ (but not along the diagonal in the plane) was also labeled 1. Breaks between pixels labeled 1 along $x$ and $y$ differentiate one group of pixels from another. In other words, the number of identified groups

defines the number of predicted particles in the plane. An isolated pixel labeled 1 was considered a particle. Note that with this procedure, overlapping masks for multiple particles are placed into the same group. The $x$ and $y$ values of identified particles were then computed as the average of the maximum and minimum extent along each direction in the group of pixels, while the diameter was estimated as the maximum extent in either direction in the group. The value of $z$ for all the particles in a plane was taken to be the value of the bin center.

Next, in the second step of matching, the list of $(x, y, z, d)$ from particles identified in $N$ planes was then used to compute the absolute distance among all combinations of pairs (excluding zero distances). Then, using the leader clustering algorithm, the particles were (potentially) assigned to clusters given a matching distance threshold. The main advantage of this approach compared with performing the pixel-grouping approach in step 1 is that using the computed particle distances allows flexibility, by means of the matching distance threshold, in how particles ultimately get assigned to clusters. However, with this approach,

particles that happen to be close in space risk being matched together and being regarded as a single particle, and the $z$ resolution will generally be lower compared with $x$ and $y$ depending on the choice of $N$. Note also that not all particles necessarily get assigned to a cluster at a given threshold. However, an extreme choice (e.g. too large) can result in all particles being regarded as a single cluster. The total number of matched particles $M$ is then the number of clusters plus the number that have been left unassigned. The $(x, y, z, d)$ value of particles that did not get assigned to a cluster are used as is. For each

cluster, the "centroid" is defined as the average value across $(x, y, z, d)$ and which represents the particles in the cluster. Once all of the cluster centroids and unassigned particles have had their position and size determined, the final list of $M$ particles is saved and the hologram is considered processed.

As is the case with synthetic holograms, the number of particles and each particle position and diameter is known precisely. The predicted number of particles $M$ will not in general be the same as the true number of particles, and will depend on the

choice of the distance threshold, the number of reconstructed planes $N$ used to obtain $z$, and the model's trained performance. In order to compare the true particle coordinates against the model predictions, the predicted particles are paired with the true particles by first computing the absolute distance among all of them, excluding distances equal to zero. Next, the predicted and true particle pair having the smallest distance as computed using $(x, y, z, d)$ for each, is taken to be a match. Both particles are then removed from further consideration. This process continues until there are no more true or predicted particles left to

pair. Thus, there can be holograms for which the number of predicted particles is short the true number, as well as holograms for which the model over-predicts the true count, and those where they are equal. The paired particles can be used to compute performance metrics such as accuracy and F1 scores, while the predicted particle numbers allow us to construct a contingency table for the holograms.

## 2.5  Model training and optimization





### 2.5.1 Hologram image transformations

The remaining methods sections describe the transformations performed on the holograms before being used as input to a model, training and optimization of a neural hologram processor as is illustrated in Figure 3(b), and the manual labeling exercises. First, we discuss several types of transformations that were performed on images before being passed into a neural network. Unlike the holograms obtained from HOLODEC, the synthetic holograms were generated to provide truth data sets to be used for training neural segmentation models. As such they did not contain any imperfections, such as background noise. As the HOLODEC examples were only processed by the standard method, which does not have perfect performance, it would be difficult to use the standard method's predicted particle positions and shapes as training labels without significant complementary manual efforts or inheriting the accuracy limitations, which are not fully known. Hence, no truth coordinates $(x, y, z, d)$ for the true particles in the HOLODEC holograms exist. Therefore, in order to produce fully accurate training data that was representative of actual HOLODEC holograms, two types of transformations were considered for application to synthetic holograms as a means of producing a trained neural network that would perform similarly on the synthetic and real holograms: (1) those that map the original input values into another range, such as centering and rescaling the values of the variables, and (2) those that perform a perturbation of the image as a means for mimicking noise signals that appear in the HOLODEC holograms.

The first type of transformations probed whether training on centered synthetic images produced higher performing models on the HOLODEC holograms, as initial investigations showed clear performance differences depending on the transformation used. The transformation types include (1) "normalizing" the hologram pixel values into the range [0, 1] by first subtracting the smallest pixel value in the image from every pixel, and then dividing every pixel by the maximum value observed, (2) "standardizing" the hologram by subtracting the mean pixel value and dividing by the standard deviation of pixel values from every pixel in the image, (3) a "symmetric" transformation that recasts the pixel values into the range [-1, 1], (4) division of all pixel values by the maximum pixel value 255, and (5) none applied. One type was selected for application to the HOLODEC and the synthetically generated holograms shown in Figure 3(b-c)(i), and another selected for application to the tiles shown in Figure 3(b)(vi) during training and optimization and in Figure 3(c)(iv) when the model is being used in operation. In both cases, the use of one of these transformations was determined during model optimization, as is discussed in Section 2.5.3.

The second type of transformations was only used during training, and was applied to tiles (if determined) just before being passed into a model, shown in Figure 3(b)(vi). If more than one was selected, they were applied one after the other. They include a transformation that adds Gaussian blur to the tile images, to help to reduce the detail contained in them by helping to remove high-frequency information. The transformation required a kernel size, which set the maximum smoothing length, and a standard deviation value to be set. Similar to the background transformation, the value of the standard deviation was selected randomly within the range between 0 and some maximum value (determined in hyperparameter optimization) while the kernel size was set to 2. A third transformation adjusts the brightness of the tile image. A brightness factor is first selected randomly between zero and a maximum value. The tile image is then multiplied by the brightness factor and then the pixel values are clipped to lie in the range bounded by 0 and 255.





A final set of data augmentation transformations, which did not involve adding noise, was always performed during model
training as a means for encouraging the model to learn different views of the same particle. They were random flips of a tile
image in either $x$ or $y$ directions, applied stochastically with probability 0.5, and applied only during model training at the step
shown in Figure 3(c)(iv). A flip in $x$ is taken to be independent of a flip along y, so that approximately one in four images will
have both $x$ and $y$ flipped. If a tile is flipped in either direction, so is the corresponding mask target. Except for the random
flips, if not stated explicitly,a transformation, as well as any relevant parameters needed to use it, were left as hyperparameters
to be optimized, as is discussed in Section 2.5.3

### 2.5.2  Training and validation metrics

Training a segmentation model to predict masks around in-focus particles required a set of input tiles and the associated masks.
The synthetic training holograms contained 50,000 particles in total. Because a particle may show up in multiple grid tiles, each
particle was randomly sampled from the subset of grid tiles that contained it. However, the model also needed to be exposed to
images that did not have any particles. This included input tiles not near any in-focus particles, as well as examples that were
close to a particle, but were out-of-focus by one z-bin increment. The latter examples depend sensitively on the choice of $N$.
For example, for small values of $N$, a particle's true $z$ may not be close to the bin center, while large values of $N$ may result in
the particle extending over several z-bins. Therefore, an additional 50,000 images of the same size as the tiles were randomly
sampled from the 100 training holograms for a given $N$, so that the number of tile examples with and without particles in focus
was balanced. Of the negative examples, approximately half were chosen to be examples where the image, if propagated one
bin in either direction along $z$, a particle would come into focus. The remaining half were randomly sampled examples that did
not necessarily contain in focus particles. The total training set of 100,000 images was then saved to disk. The same procedure
was performed on the validation and testing holograms, which produced validation and testing data set sizes of 10,000 images
in total.

Next, fixed-sized subsets of input images and output masks were selected from the training images to create input batches to
the neural network. If instructed to do so, any of the transformations described above were then applied to each image in the
batch (Figure 3(b)(vi)). The batches were passed through the model to obtain a mask prediction for each image in the batch.
The predicted masks were compared against the true data with a loss function that computes, for example, the mean-absolute
error (see Figure 3(b)(viiii)). Using the computed loss value, the weights of the model were updated using gradient descent
through back-propagation (Rumelhart et al., 1986). This process was repeated until a fixed number of training batches, which
is the total number of training samples divided by the batch size, are passed through the model once, and is referred to as one
epoch of model training. The training data was randomly shuffled once all examples passed through the model. The choice of
the training loss was left as a choice to be optimized (see the next section).

After each epoch, the model was placed into evaluation mode, which disabled any stochastic elements, and used to make
predictions with the validation set as inputs. The loss was computed for each example in the validation set, and then averaged
to produce a single value for the set. The procedure of training for one epoch then computing a validation loss was repeated for
a prescribed number of epochs. If the validation loss value stopped improving after 3 epochs, the learning rate was reduced by





a factor of 10. Furthermore, if the validation loss had not improved after six epochs, model training was taken to be completed, otherwise model training was terminated after 200 epochs.

The validation loss was taken to be the (smoothed) dice coefficient, which is given by

$$\text{smoothed dice}(x,y) = \frac{2\sum_j x_j y_j + 1}{\sum_j x_j + \sum_j y_j + 1} \tag{6}$$

and was computed by flattening the 2D input images into 1D arrays. The sum in the numerator is the element-wise product of predictions $x$ and labels $y$. The raw outputs from the model, which lie within the range [0, 1], are used rather than the (integer) binary label, which required casting the model outputs to integer values which would not allow the product calculation
(intersection) to be differentiated. Both the numerator and denominator contain a smoothing factor (+1) to help prevent division by zero if the inputs contain only null values.

### 2.5.3    Model optimization

As noted in earlier sections, at different stages in training a neural network model there are hyperparameters that need to be set that affect the performance outcome of a trained model. Both the type of segmentation model and the layers from a
separate pre-trained model that were used in the composition of the segmentation encoder model (if the segmentation model was composed of encoder-decoder components), were taken to be hyperparameter choices to be optimized along with others. The choice of segmentation model was made from a fixed set of available models, as well as the pre-trained encoder model, but the choice of both is only selected if the pre-trained layers can be used in the encoder-head of the segmentation model (see Section A2 for a full list of segmentation and encoder models that we evaluated).
Other hyperparameters include selection of the training loss function from a fixed set that included Eq. 6 as well as others (see Section A2 for the full list of losses used), the starting value of the learning rate, the transformation type applied to full-sized images before performing wave-propagation shown in Figures 3(b-c)(i) (none, standard, or normalization) and to the tiles at the step shown in Figure 3(b)(vi) (none, standard, normalization, symmetric, division by 255). Lastly, the parameters involved in the transformations to input tiles that introduce background noise, Gaussian blurring and contrast adjustment, applied at the
step shown in Figure 3(b)(vi), were chosen to be optimized.

We used the package Earth Computing Hyperparameter Optimization (ECHO) developed by the authors at NCAR (Schreck and Gagne, 2021), to perform optimization using Eq. 6 computed with the synthetic validation data set by varying hyperparameters. The optimization begins with a randomly selected set of choices for the above mentioned hyperparameters, and is used to train a model as discussed above. The largest validation loss computed using Eq. 6 observed during training was used
to score the performance of the hyperparameter set. Random selection of hyperparameter sets was repeated 200 times. Then, a Bayesian strategy leveraged the validation loss values from previous trials to make the next hyperparameter selection that aimed to maximize Eq. 6. This was repeated for several hundred more trails until the algorithm was observed to approximately converge. Finally, the best performing set of hyperparameters was then selected and one final training was performed to obtain the optimized model weights.

### 2.5.4 Manual labels for a set of HOLODEC holograms

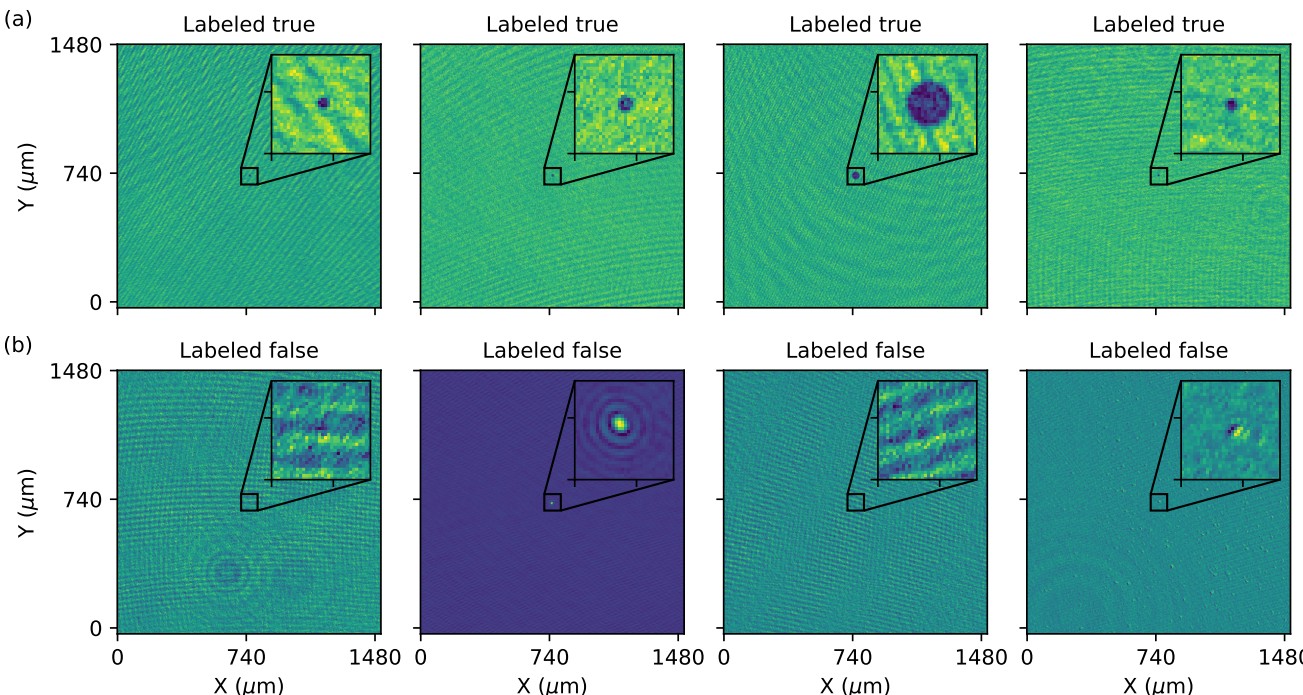

**Figure 4.** Examples of manually labeled, wave-propagated holograms. The top row shows cases where the particle in the image was determined to be in focus (labeled true), while the bottom row illustrates cases where no particle was in focus. The second and fourth examples on the bottom illustrate reflection, and what was determined to be an artefact, respectively. The insets highlight the center of each image.

The synthetic holograms were simulated to produce a truth data set for use in training neural network segmentation models, while the HOLODEC examples processed by the standard method did not necessarily represent the true $(x, y, z, d)$ values for the particles in a hologram. As the synthetic holograms were all generated with simulations absent any noise, optimization of models converged toward parameters that produced noise-free or low-noise transformations being applied during model training. Therefore, some examples of HOLODEC holograms that contained real-world imperfections were required during training and optimization, to enable effective parameterization of hyperparameters associated with the noise transformations. Further, we also needed to know the performance of the standard method on the HOLODEC holograms considered, so that a fair comparison between a neural network model and the standard method could be made.

The outcomes from the standard method on 10 HOLODEC holograms were used to select examples where a particle was predicted to be in focus. Using the predicted particle location along $z$, the hologram was wave-propagated to the nearest bin center with $N = 1,000$. Then, using the predicted center, a sub-sample of the propagated hologram was selected, where the particle was positioned at the image center, and which was the same size as the grid tiles used above (examples where the





particle was close to an edge were padded so that the particle remained at the center of the image). A neural network optimized on the synthetic holograms that utilized noise transformations was also used to provide additional examples (illustrated in
Figure 3(b)(v), see below for more details). In total, 1202 images were produced for consideration where some were identified as in focus by the standard method, some were identified by the neural net, and in some, but not all cases, these predictions overlapped. These examples are referred to below as the validation set of HOLODEC holograms.

Each example in the validation set was then labeled as containing an in-focus particle (true), or not (false), by the authors. Information about which method identified a particle was withheld. Each image label was also assigned a confidence score
by a reviewer, ranging from 1 (no confidence) to 5 (most confident). The final label assigned to an image was determined by computing a weighted average for the label, and then labeling 1 if the score was larger than 0.5 and 0 otherwise. Several examples of hand-labeled images are shown in Figure 4, with positive examples shown in the top row and negative examples shown in the bottom row. A trained neural network model was selected for use due to its high rate of false positive predictions on the HOLODEC holograms, to ensure that both positive (mainly, but not exclusively, from the standard method) and negative
examples (mainly, but not exclusively, from a trained model) would be sampled. In total, 367 examples were labeled 1 compared to 835 labeled 0.

### 2.5.5  Optimization with synthetic and HOLODEC holograms

Model optimization was performed with the noise transformations being used during the training runs. After the end of every training epoch, in addition to computing the dice loss on the validation set of synthetic images for mask predictions, the model
was used to make binary predictions on the validation HOLODEC holograms. If a mask was predicted by the model it was labeled 1 otherwise it was labeled 0. This step is shown schematically in Figure 3(b)(viii). The average dice coefficient was then computed for the manually labeled set according to Eq. 6 and added to that computed for the mask prediction task for the synthetic images (this step is shown in Figure 3(b)(viii-x)). Similarly, during hyperparameter optimization, the objective metric was taken to be the sum of the computed dice coefficients.

Lastly, a second round of manual evaluation focused on all positive predictions by the standard method, and those by a trained model with optimized hyperparameters which utilized both the synthetic holograms and the validation HOLODEC examples as a means for optimizing the parameters associated with noise added during training. The labeling was performed with 10 additional HOLODEC holograms not used in the first round, and which is referred to as the test set of HOLODEC holograms. The test set was used for computing and comparing the performance of the standard method and the neural network model
just after the clustering step as judged by the four reviewers, the steps followed are shown schematically in Figure 3(c)(vii-x). In total 1,154 examples were considered by the reviewers, which had 890 examples labeled 1 and 264 labeled 0. The same examples were also used to estimate the approximate $(x, y, z, d)$ coordinates for in-focus particles. For example, when both the standard method and the neural model predicted a particle to be in-focus, the $(x, y, z, d)$ assigned to the in-focus particle was taken to be the average of the two predictions. In remaining cases, $(x, y, z, d)$ was selected to be the output of the model that
predicted the particle to be in-focus.





| Metric | F1 | | AUC | | POD | | FAR | | CSI | |
|---|---|---|---|---|---|---|---|---|---|---|
| | Synth | HOLO | Synth | HOLO | Synth | HOLO | Synth | HOLO | Synth | HOLO |
| $N_S = 1,000$ | 0.977 | 0.279 | 0.993 | 0.372 | 0.982 | 0.186 | 0.029 | 0.432 | 0.954 | 0.163 |
| $N_{SH} = 100$ | 0.911 | 0.786 | 0.984 | 0.756 | 0.956 | 0.794 | 0.129 | 0.102 | 0.837 | 0.728 |
| $N_{SH} = 1,000$ | 0.962 | 0.881 | 0.985 | 0.807 | 0.964 | 0.953 | 0.040 | 0.102 | 0.927 | 0.860 |
| $N_{SH} = 5,000$ | 0.961 | 0.885 | 0.990 | 0.799 | 0.970 | 0.983 | 0.047 | 0.112 | 0.926 | 0.875 |
| $N_{SH} = 10,000$ | 0.962 | 0.841 | 0.983 | 0.803 | 0.963 | 0.870 | 0.040 | 0.089 | 0.926 | 0.802 |
| $N_{SH} = 48,648$ | 0.962 | 0.888 | 0.991 | 0.799 | 0.971 | 0.987 | 0.046 | 0.112 | 0.927 | 0.878 |

**Table 1.** Several metrics are listed for each model and were computed on the synthetic (Synth) and HOLODEC (HOLO) test data sets, for mask and particle detection (binary) predictions, respectively. The values of POD and CSI reported for mask prediction used the probability threshold which maximized CSI.

## 3  Results

### 3.1  Model performance on the synthetic and HOLODEC test tiles

We investigated six models to probe the performance dependence on the choice of $N$ as well as the noise introduced during training. The first model was optimized on synthetic holograms as described in Section 2.5.3 using $N_S = 1,000$ bins along the
z direction. The subscript 'S' refers to the synthetic data set used. The remaining models used different resolutions along $z$, which were $N_{SH} = 100, 1,000, 5,000, 10,000$, and 46,648. As described in Section 2.5.5, the $N_{SH}$ models were trained on synthetic holograms that are corrupted by noise processes. We optimized the corruption in hyperparameter optimization by utilizing the manually labeled HOLODEC examples, hence the subscript 'SH', which is used to differentiates these models from the baseline 'S' model.

Table 1 lists several binary performance metrics computed for the models on the synthetic and HOLODEC test data sets containing the randomly sampled tiles, as they were not used in any way during training and optimization. These metrics measure how well each model performed at predicting masks around in-focus particles in synthetic tiles, and at detecting in-focus particles in HOLODEC tiles. The synthetic test set contained 10,000 tiles and was balanced between tiles containing in-focus particles and no in-focus particles, while the HOLODEC test set contained 1,154 example tiles and contained about
3.4 examples of in-focus particles for every example containing no particle. Figure A1 also plots the training performance as a function of epochs on the validation sets of synthetic and HOLODEC images (see Section A2 for further training and optimization details and results).

Table 1 shows that all of the trained models performed well on the synthetic test tiles, where the F1 score, area-under-the-ROC-curve (AUC), probability of detection (POD), and maximum critical success index (CSI) (Wilks, 2011) were all greater than 0.9 indicating strong mask prediction performance. With the exception of $N_{SH}$=100, the false alarm ratio (FAR) was less than 5%. Additionally, the $N_S = 1,000$ model, which was trained exclusively on noise-free synthetic images, generally had



| Metric | Ave. # | Rate | F1 | AUC | POD | FAR | CSI |
|---|---|---|---|---|---|---|---|
| $N_S = 1,000$ | 1245 | 1.24 | 0.624 | 0.985 | 0.920 | 0.528 | 0.453 |
| $N_{SH} = 100$ | 395 | 3.95 | 0.883 | 0.921 | 0.841 | 0.071 | 0.791 |
| $N_{SH} = 1,000$ | 1276 | 1.28 | 0.558 | 0.970 | 0.870 | 0.589 | 0.387 |
| $N_{SH} = 5,000$ | 7297 | 1.46 | 0.130 | 0.976 | 0.892 | 0.985 | 0.069 |
| $N_{SH} = 10,000$ | 11125 | 1.11 | 0.084 | 0.961 | 0.846 | 0.956 | 0.044 |
| $N_{SH} = 48,648$ | 64163 | 1.32 | 0.016 | 0.952 | 0.802 | 0.992 | 0.008 |

**Table 2.** The average number of predicted particles divided by the number of planes $N$, and the computed binary metrics for the mask prediction task are listed for each model. The metrics were computed using the 10 test holograms, where each hologram contained 500 particles.

higher performance scores on the synthetic test tiles compared to the other models. The effect on performance from added noise during training is clearly seen by the modest drops in the computed metrics for the $N_{SH}$ models. On the other hand, the binary performance on the HOLODEC test tiles went up significantly for the $N_{SH}$ models, and which peaked with the $N_{SH} =$ 485  48,648 model, but 1,000, 5,000, and 10,000 all had comparable performance.

Overall, the noise added to synthetic tiles during training, and the manually labeled HOLODEC tiles that were used to influence the optimization of the neural network weights, resulted in trained models having lower performance on synthetic holdout tiles, but higher performance on the holdout HOLODEC tiles. Furthermore, the models optimized with labeled HOLODEC tiles clearly outperformed those trained with just synthetic examples. The table also shows that models with $N_{SH}$ greater than 490  or equal to 1,000 all outperformed the $N_{SH} = 100$ model, which clearly had worst performance on the HOLODEC test tiles for models trained with noise, but still it outperformed the model trained absent any noise by a wide margin.

### 3.2    Reassembled model performance on test synthetic holograms

With the models listed above, the values for $(x, y, z, d)$ were determined for the particles in each of the 10 test synthetic holograms according to the steps illustrated in Figure 3(c) that involved reassembly, matching, and pairing predicted particles 495  with true ones. Table 2 shows the same performance metrics for the predicted masks for the 10 test holograms as in Table 1, here computed across all planes as a function of $N$, for the reassembly step illustrated in Figure 3(c). That is, before clustering but after the predicted probabilities for the grid tiles have been combined to create hologram-sized predictions for each reconstructed plane, and concatenated along $z$ to create a 3D prediction array.

Table 2 shows that the $N_{SH} = 100$ model produced the highest values on AUC, POD, maximum CSI and the lowest on 500  FAR, while the $N_{SH} = 48,648$ model was the worst overall performer. These three metrics are clearly correlated with the total average number of particles predicted per hologram, which grows large as the number of planes used along $z$ increased. In particular, the larger the number of planes, the lower the AUC, POD, CSI, and higher POD. Only the $N_{SH} = 100$ model under

**Figure 5.** Three successive planes are shown in (a-c). Each row illustrates the tile image at $z$, the center of tile image zoomed-in, the prediction by the standard method, and the mask prediction by the $N_{SH} = 1,000$ neural network model.

predicted the true number (500), while the other models predicted numbers that were proportional to $N$ and which had a similar particle prediction rate for a plane, as measured by the total number predicted divided by $N$ in Table 2.

The dependency of POD on $N$ relative to FAR, which did not exhibit as strong a dependence on $N$ by comparison, suggests that the models generally predicted a mask around a particle when it was actually in focus, while at the same time increasingly over-predicting the same particle in successive planes as $N$ increased. Figure 5 illustrates a model predicting a mask around a large particle, but over three successive planes, as well as the plane identified by the standard method as that which was closest to the in-focus particle. The standard method and the neural model agreed that the image reconstructed at the z-plane shown in (b) contains an in-focus particle. However, the neural network does not differentiate from the three examples, as viewed from the front and backside of the particle, as seen in Figure 5(a) and Figure 5(c), respectively. The example shows $N_{SH} =$



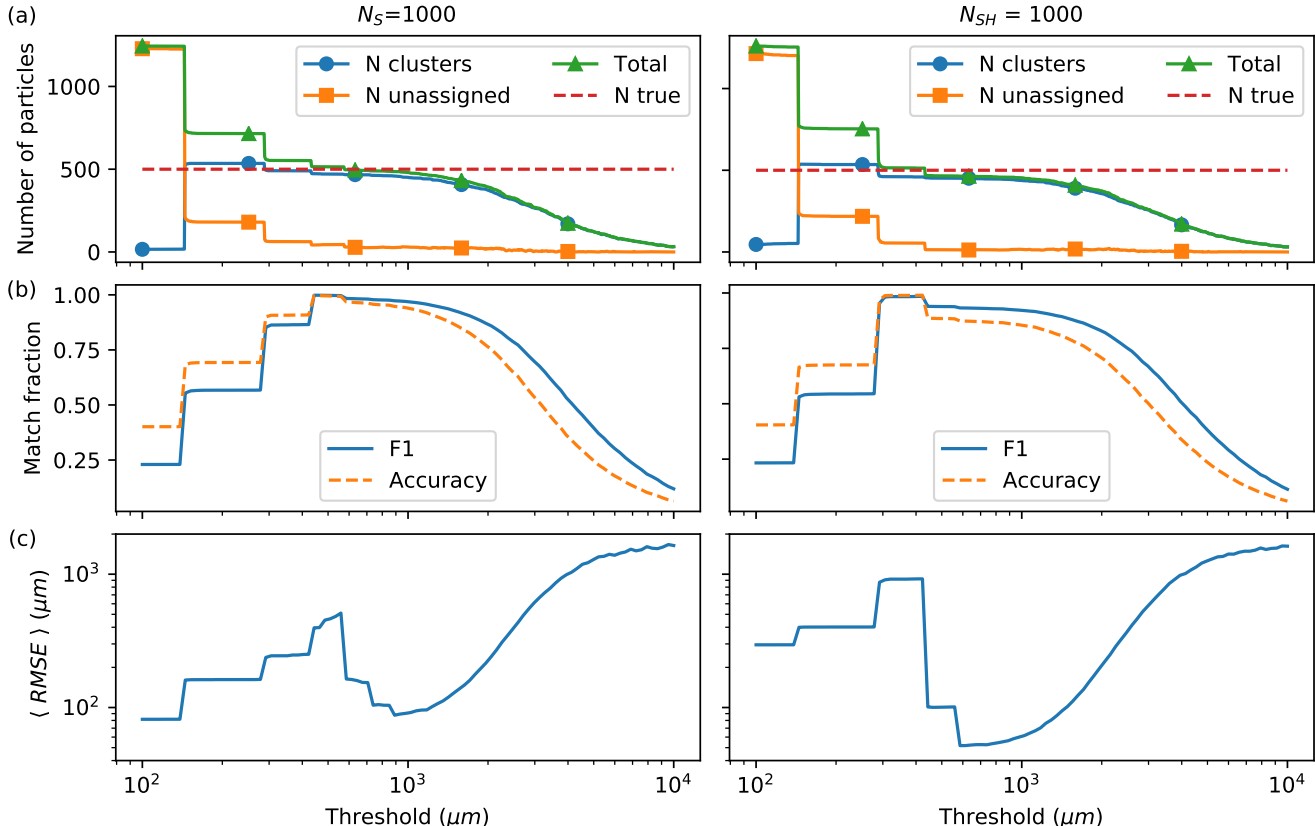

**Figure 6.** In (a-c), the number of particles, the matched F1 and accuracy metrics, and the average RMSE, computed for predicted particles paired with true particles, are all shown versus the match distance threshold, respectively. The left and right columns show the results for the noise-free $N_S = 1,000$ model, and the noise-optimized $N_{SH} = 1,000$ model, respectively.

1,000, where larger particles primarily are over-predicted along the $z$ coordinate. However, as $N$ increased, smaller particles also appeared to be in focus in greater numbers of reconstructed planes.

### 3.3 Coordinate and diameter performance on synthetic holograms

The 3D prediction array for each test synthetic hologram were clustered using a range of match distance thresholds (referred to as thresholds). Figure 6(a) shows how the choice of threshold determines the number of predicted particles that get assigned to clusters (circles) and also those that do not (squares). The total number of particles (triangles) is the number of clusters plus the number of unassigned particles for a given threshold. In Figure 6(b), the match accuracy and F1 metrics were computed using the total number of predicted particles and the total number of true particles in the hologram (dashed lines in Figure 6(a)).

The match accuracy measures the fraction of predicted particles having been paired against the true number in each hologram. When the threshold value is small in the figure, the match accuracy is higher than the match F1 score because there is an excess





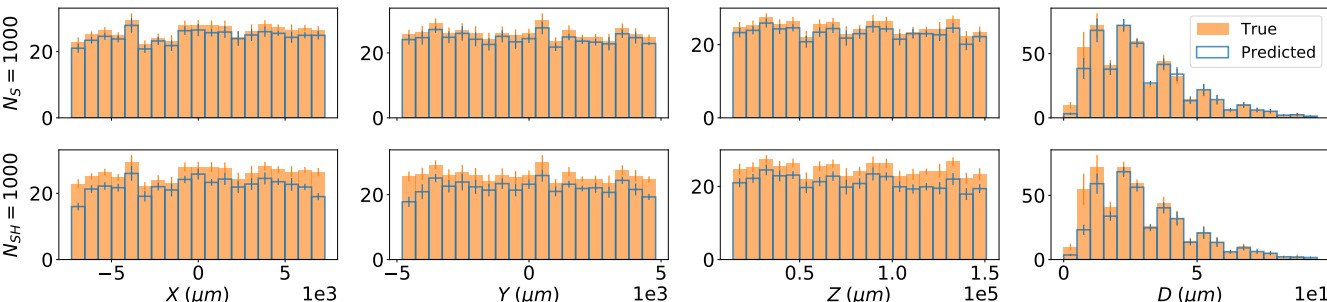

**Figure 7.** Histograms for the particle coordinates $(x, y, z, d)$ are shown in each column compare the models predictions (blue, outlined boxes) against the true values (orange, solid boxes), computed using the 10 test (synthetic) holograms. The results for models $N_S = 1,000$ and $N_{SH}$ = 1,000 are shown in (a) and (b) respectively. The value of each bin and the error bar were computed by taking the mean and standard deviation across the 10 holograms.

number of predicted particles. At larger threshold values there are fewer particles compared with the total true number, hence the match F1 score is higher than the match accuracy. Figure 6(c) shows the computed RMSE for the matched particles paired with true ones versus the threshold value for the $N_S = 1,000$ and $N_{SH} = 1,000$ models.

At small thresholds, most of the predicted particles have not been clustered for either model. Hence, there is an excess of predicted particles compared with the true number. As the threshold increases to approximately the spatial separation of the reconstructed planes, which is $144 \, \mu m$ for $N = 1,000$, many particles have joined clusters and the total number of clusters plus unassigned particles quickly drops close to the number of true particles. Then a flattening of the curves occurs over a small range of thresholds in Figure 6(a), approximately centered around $1000 \, \mu m$. Figures 6(b-c) also show that in this regime of

threshold values, the highest values of match F1 and accuracy were observed, and the lowest computed RMSE values between predicted and true particle pairs, respectively. However, once the threshold becomes large relative to the size of the system, most of the particles were grouped into a few clusters.

The lowest average RMSE was always observed at a threshold value which resulted in fewer particles predicted than the true number. In fact, the $N_S = 1,000$ model had the lowest computed RMSE at $84 \, \mu m$ when 93% of predicted particles were paired

with true particles (threshold of $890.2 \, \mu m$), while it was $50.7 \, \mu m$ for the $N_{SH} = 1,000$ model when the match accuracy was at 85% (threshold of $585.7 \, \mu m$), and which is close to the depth of field for the instrument of $57 \, \mu m$. Overall, the noise-free $N_S$ = 1,000 model had a higher match accuracy and F1 over the range of thresholds compared with to the $N_{SH} = 1,000$ model, consistent with Tables 1 and 2, with generally larger computed RMSE between the paired predicted and true particles.

The reason for the higher RMSE in the noise-free model $N_S$ compared to $N_{SH}$ models is due to the higher match accuracy.

This can be observed in Figure 7, which shows histograms for the predicted and true $(x, y, z, d)$ coordinates for the particles in the 10 test synthetic holograms using a threshold of $1000 \, \mu m$ for the $N_S = 1,000$ and $N_{SH} = 1,000$ models, respectively. Table 3 also shows the same mask prediction performance metrics for the two models after matching and pairing at the same threshold.





| Metric | Ave. # | F1 | AUC | POD | FAR | CSI |
|---|---|---|---|---|---|---|
| $N_S = 1{,}000$ | 465 | 0.922 | 0.964 | 0.929 | 0.085 | 0.855 |
| $N_{SH} = 100$ | 372 | 0.869 | 0.906 | 0.813 | 0.066 | 0.769 |
| $N_{SH} = 1{,}000$ | 423 | 0.911 | 0.950 | 0.900 | 0.078 | 0.837 |
| $N_{SH} = 5{,}000$ | 437 | 0.433 | 0.693 | 0.386 | 0.506 | 0.277 |
| $N_{SH} = 10{,}000$ | 417 | 0.403 | 0.663 | 0.323 | 0.470 | 0.253 |
| $N_{SH} = 48{,}648$ | 409 | 0.420 | 0.713 | 0.427 | 0.587 | 0.266 |

**Table 3.** The same mask prediction performance metrics shown in Table 2 are listed for predicted particles paired to true particles averaged over the 10 test synthetic holograms. The match distance threshold used was $1000\,\mu m$.

The strong overlap in all of the histograms in Figure 7, and the high mask prediction performance listed in Table 3, both
show that the clustering procedure was mostly successful at assigning particles, such as those illustrated in Figure 5, into the same cluster, for both models. However, clearly the model trained without noise (top row) shows greater overlap between predicted and true particles, due to the greater number of predicted particles, and generally had slightly better mask-prediction performance. Comparing the two histograms for the predicted particle diameters indicates that adding noise to the synthetic images during training resulted in a lower ability by the model to identify and predict the coordinates of the smaller particles.
The predicted distributions for $x$ and $y$ suggest that noise may have also lowered the model performance near the edges of a plane. But the generally higher observed RMSE seen in Figure 6(c) for the $N_S = 1{,}000$ model indicates that these particles were also the hardest to predict and cluster precisely in 3D when noise was absent during training. As a result, the $N_S = 1{,}000$ model had a marginally higher FAR compared to $N_{SH} = 1{,}000$.

### 3.4 Performance dependency on $N$

Lastly, the performance dependence on the choice of $N$ is characterized at the $1000\,\mu m$ threshold. Figure 8 shows the same performance metrics as in Figure 6(b-c) for the four $N_{SH}$ models. Table 3 compares their mask-prediction performance for paired particles and Table A2 compares their average coordinate and RMSE predictions.

Overall, the performance for $N > 1{,}000$ for varying threshold values was found to be either comparable or lower relative to 1,000. For example, the RMSE computed for $N_{SH} = 48{,}648$ improved on $N_{SH} = 1{,}000$ by $1\,\mu m$, but with a lower match
accuracy of 82% compared to 85%. The maximum CSI for the paired particles was also lower for the $N_{SH} = 48{,}648$ model. Similarly, the $N_{SH} = 10{,}000$ model had a slightly better estimate of $z$ and the RMSE (Table A2), but predicted fewer total particles that were less aligned in 3D relative to 1,000 (see Table 3). The lack of improvement observed with increasing $N$ is mainly due to the problem of models over-predicting the same particle in multiple planes, as illustrated in Figure 5.

Figure 9 shows the computed histograms for the models for each coordinate $(x, y, z, d)$ along with the true particle his-
tograms. Even though no improvement was observed for $N$ larger than 1,000, overall all of the models showed strong overlap relative to the true distribution for each coordinate, as the match accuracy was higher than 80% in each case except for $N_{SH} =$

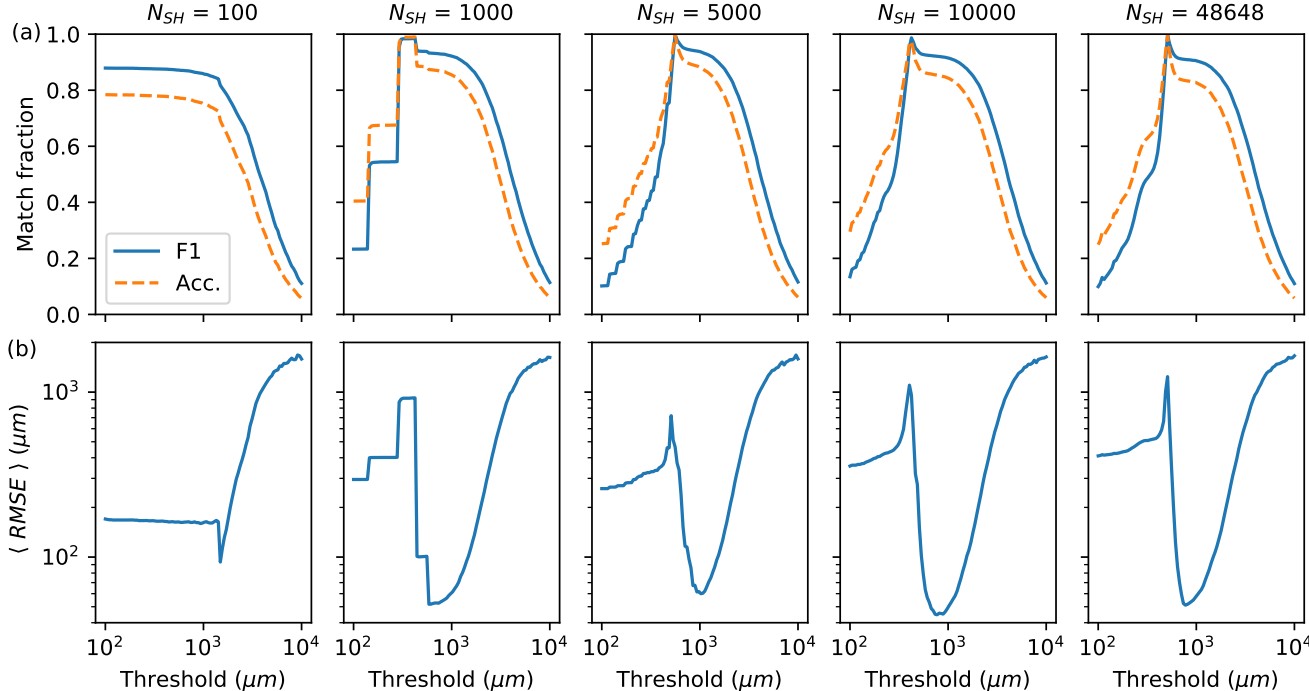

**Figure 8.** (a) The match accuracy and F1, and (b) the average RMSE for paired particles, are each plotted versus the match distance threshold. The columns show the results for the five models trained and optimized on synthetic and HOLODEC images with the number of planes used for reconstruction increasing from left to right.

where it was 75%. The increased difficulty in predicting edge particles, as caused by the noise introduced during training, is apparent for each value of $N$, while the histograms for particle diameters show that increasing $N$ from 1,000 decreases the overlap between true and predicted bins corresponding with larger diameter sizes. The predicted distributions for $z$ may also
indicate that larger values of $N$ slightly lowers the accuracy at larger values of $z$, relative to $N = 100$.

### 3.5 The standard method and $N_{SH}$ performance on HOLODEC holograms

Finally in this section, the performance of the standard method is compared against that for the $N_{SH} = 1,000$ model using a matching threshold of $1000\,\mu m$. The test set of manually labeled HOLODEC holograms was used to estimate the performance of particle detection after matching, and the approximate error in the predicted $(x, y, z, d)$ coordinates. All particles labeled true
could be assigned coordinates because the examples selected for the second round of manual examination were all positive predictions from the standard method and the $N_{SH} = 1,000$ model. The standard method only began searching for particles at $z = 20.000\,\mu m$ while we selected $z = 14.300\,\mu m$ for the $N_{SH} = 1,000$ model. Below, reported performance metrics comparing the two used only the examples with z-values which were examined by both approaches.

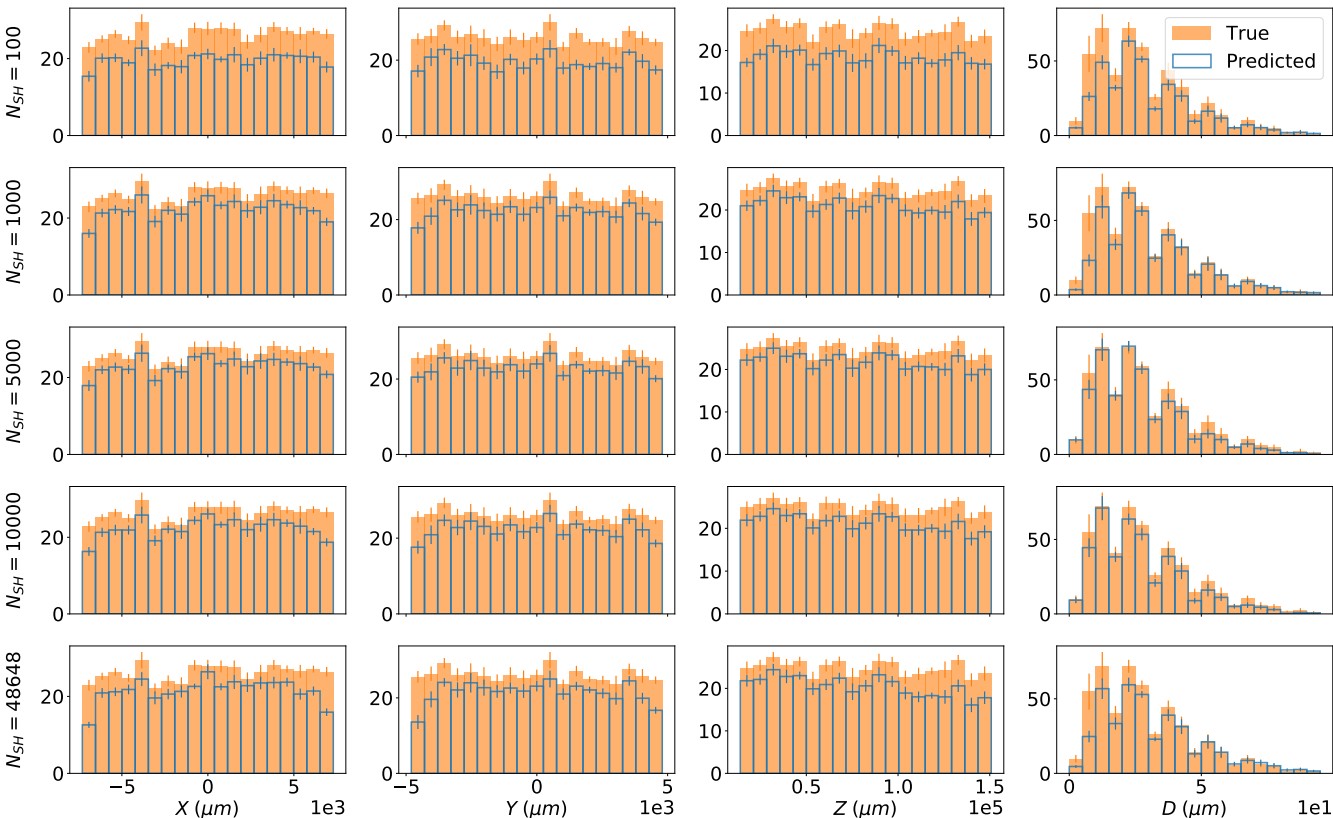

**Figure 9.** Histograms for the particle coordinates and diameter are shown in each column compare the models predictions (blue, outlined boxes) against the true values (orange, solid boxes), computed using the 10 test holograms. The rows show the results for models $N_{SH} = 100$, 1000, 10,000, and 48,648, respectively. The value of each bin and the error bar were computed by taking the mean and standard deviation across the 10 holograms. The match distance threshold was $1000\,\mu m$.

Table 4 shows several performance metrics comparing the particle-detection ability of the standard method and the $N_{SH} =$ 1,000 model. Of the 1,109 examples considered in the comparison, 847 were manually determined to contain at least one in-focus particle, with the standard method and the $N_{SH} = 1,000$ model each identifying 771 and 839 true particles, respectively. The table clearly shows that the $N_{SH} = 1,000$ model had higher average performance on all computed metrics, in particular besting the standard method by more than 20 percent on the F1 score (87.8% versus 66.8%). Inspection of individual hologram performances revealed that the standard method struggled with hologram examples containing zero particles, by predicting significant numbers of false-positive particles, while it performed much better with hologram examples containing hundreds of particles. By contrast, the neural network performed very well on all the examples containing no particles, and had higher performance on the other holograms with the exception of example 13, where the the standard method slightly out performed the neural network.





| Id | N examples | N true | F1 | | AUC | | POD | | FAR | | CSI | |
|---|---|---|---|---|---|---|---|---|---|---|---|---|
| | | | Standard | N. N. | Standard | N. N. | Standard | N. N. | Standard | N. N. | Standard | N. N. |
| 10 | 253 | 215 | 0.755 | 0.9 | 0.414 | 0.923 | 0.898 | 0.902 | 0.154 | 0.03 | 0.772 | 0.878 |
| 11 | 89 | 70 | 0.735 | 0.885 | 0.588 | 0.952 | 0.886 | 0.943 | 0.184 | 0.083 | 0.738 | 0.868 |
| 12 | 267 | 243 | 0.828 | 0.888 | 0.753 | 0.912 | 0.835 | 0.918 | 0.065 | 0.051 | 0.79 | 0.875 |
| 13 | 179 | 158 | 0.846 | 0.836 | 0.761 | 0.915 | 0.873 | 0.899 | 0.068 | 0.09 | 0.821 | 0.826 |
| 14 | 149 | 127 | 0.772 | 0.823 | 0.718 | 0.915 | 0.78 | 0.953 | 0.092 | 0.123 | 0.723 | 0.84 |
| 15 | 41 | 34 | 0.7 | 0.756 | 0.643 | 0.874 | 0.647 | 0.853 | 0.083 | 0.147 | 0.611 | 0.744 |
| 16 | 30 | 0 | 0.125 | 0.966 | - | - | 0.0 | 0.0 | 1.0 | 1.0 | 0.0 | 0.0 |
| 17 | 48 | 0 | 0.0 | 1.0 | - | - | 0.0 | 0.0 | 1.0 | 0.0 | 0.0 | 0.0 |
| 18 | 23 | 0 | 0.0 | 0.955 | - | - | 0.0 | 0.0 | 1.0 | 1.0 | 0.0 | 0.0 |
| 19 | 30 | 0 | 0.0 | 1.0 | - | - | 0.0 | 0.0 | 1.0 | 0.0 | 0.0 | 0.0 |
| | **1109** | **847** | **0.668** | **0.878** | **0.442** | **0.930** | **0.847** | **0.915** | **0.230** | **0.076** | **0.676** | **0.851** |

**Table 4.** Table of metrics for the standard method and the $N_{SH} = 1,000$ neural network model (denoted N. N.) computed using the test set of manually labeled HOLODEC holograms. The number of examples equals the total number predicted by both the standard method and the neural network, while the true number was determined by manual examination. The confidence scores assigned to the manual labels by the reviewers were used in the AUC calculations.

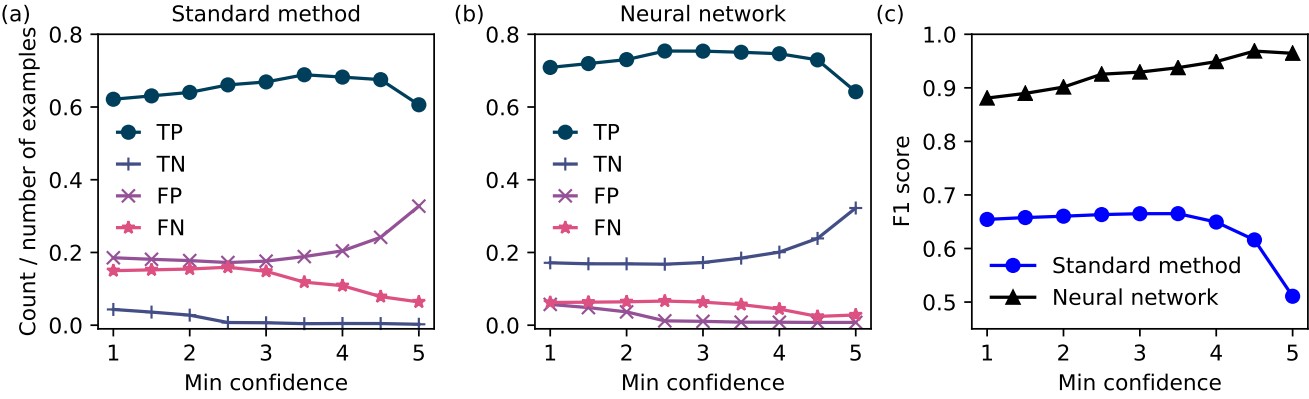

**Figure 10.** The computed number of true positives (TP, circles), true negatives (TN, plus), false positives (FP, cross), and false negatives (FN, star) versus the minimum confidence score, for the standard method and the neural network model in (a) and (b), respectively. The F1 score versus confidence is shown in (c) for the standard method (circles) and the neural model (triangles). Examples having confidence lower than the minimum were not included in the calculation.

As noted, the manual labels were also assigned confidence scores by each reviewer. Examples ranged from clearly containing
in-focus particles (average score = 5) to those which the reviewers essentially could not determine if the prediction was a



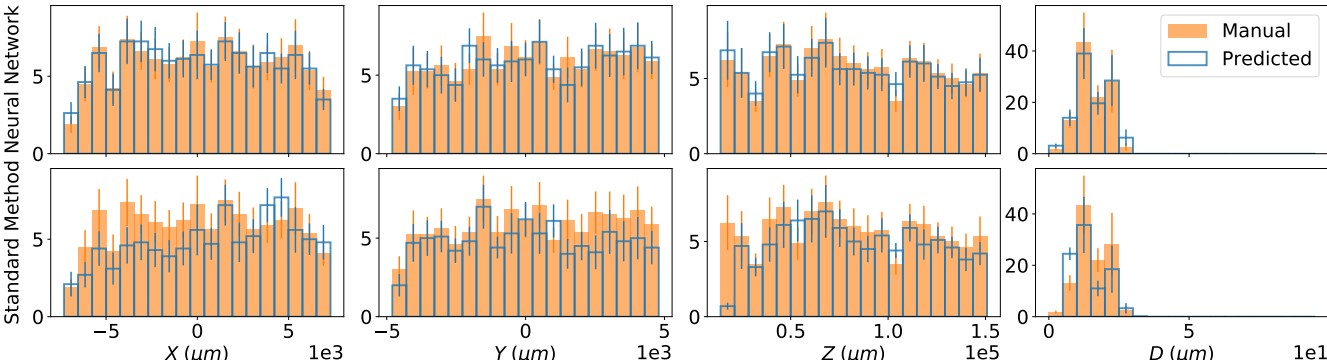

**Figure 11.** Histograms for $x$, $y$, $z$, and $d$ are shown in each column, computed using the 10 testing HOLODEC holograms. The results for the standard method and the $N_{SH} = 1000$ model are shown in (a) and (b), respectively. The value of each bin and the error bar were computed by taking the mean and standard deviation across the 10 holograms.

particle (average score = 1). Figure 10 shows the performance of the confusion matrix and F1 score for the standard method and the $N_{SH} = 1,000$ model, versus the average confidence of the manual determination. The metrics were computed on the subset of images which had average confidence at least as large as the values shown in the x-axes in Figure 10.

The true positive rate was higher for the $N_{SH} = 1,000$ model across the confidence scores. The standard method had a larger
false positive rate that increased with confidence, while the $N_{SH} = 1,000$ model had a higher true negative rate that increased with confidence. As seen in Figure 10(c), these observations translated into an increasing F1 score with increasing average confidence for the $N_{SH} = 1,000$ model, while it remained flat and then decreased for the standard method.

Figure 11 compares the estimated distributions for $(x, y, z, d)$ for the manually determined particles in HOLODEC test holograms against those predicted by the standard method, and $N_{SH} = 1,000$ model using a threshold of $1000\,\mu\text{m}$ (unpaired
distributions are shown in Figure A2). As described in Section 2.5.5, the estimates were obtained from the standard method and/or the neural network for the images labeled to contain an in-focus particle. Figure 12 illustrates the 3D predictions by the standard method and the neural model relative to the manually determined particles for HOLODEC example 10.

The high performance of the $N_{SH} = 1,000$ model is reflected by the strong overlap between predicted and true histogram for each coordinate, while the overlap is clearly lower for the standard method. This can also be seen for the example illustrated
in Figure 12 by comparing the top and bottom panels for either approach. Aside from the clear discrepancy at small $z$ for the standard method (included are the particles manually labeled in focus that were below the search window of the standard method, but were found by the neural network), there was not any clear performance bias observed at the edges of a hologram, along $z$, or with $d$ for either model (however, for unpaired predictions the standard method may be biased toward one side of a hologram. See Figure A2). The average absolute error between the paired true and predicted particles for $(x, y, z, d)$, and
the RMSE, were mainly comparable between the two models, with the neural network having slightly lower RMSE ($39.5\,\mu\text{m}$ versus $45.1\,\mu\text{m}$) while the standard method performed slightly better on diameter estimation (see Table A3).



**(a)(i) True particles**   **(b)(i) Standard method predictions**   **(c)(i) Neural network predictions**

**(a)(ii) True particles**   **(b)(ii) Standard method matches**   **(c)(ii) Neural network matches**

**Figure 12.** The particles in hologram 10 from the HOLODEC test set are plotted in 3D for (a) the true particles as determined by manual evaluation, (b) the standard method predictions, and (c) the neural network predictions. In (b-c)(i) all predictions are shown for either approach. In (b-c)(ii) only pairs between true particles and those predicted by the standard method or the neural network, respectively, are shown.

Lastly, Figure 13(a) and (b) illustrate examples where the standard method and the neural network model made incorrect predictions, respectively. The false positive examples in Figure 13(a) show the standard method detecting a reflection, and some sort of "artefact" as particles, respectively. Examples like these represented nearly all of the false positive predictions in 615 the holograms that were determined to contain zero particles, as well as in many of the higher density holograms (see Table 4). Additionally, the manual evaluation for many of these examples were typically of high confidence, as they were easy to spot by eye, and were mainly the reason for the increasing false-positive and true-negative rates with confidence, as was seen in Figures 10(a) and (b), respectively, as the neural network model correctly predicted no in-focus particle for many of them. False



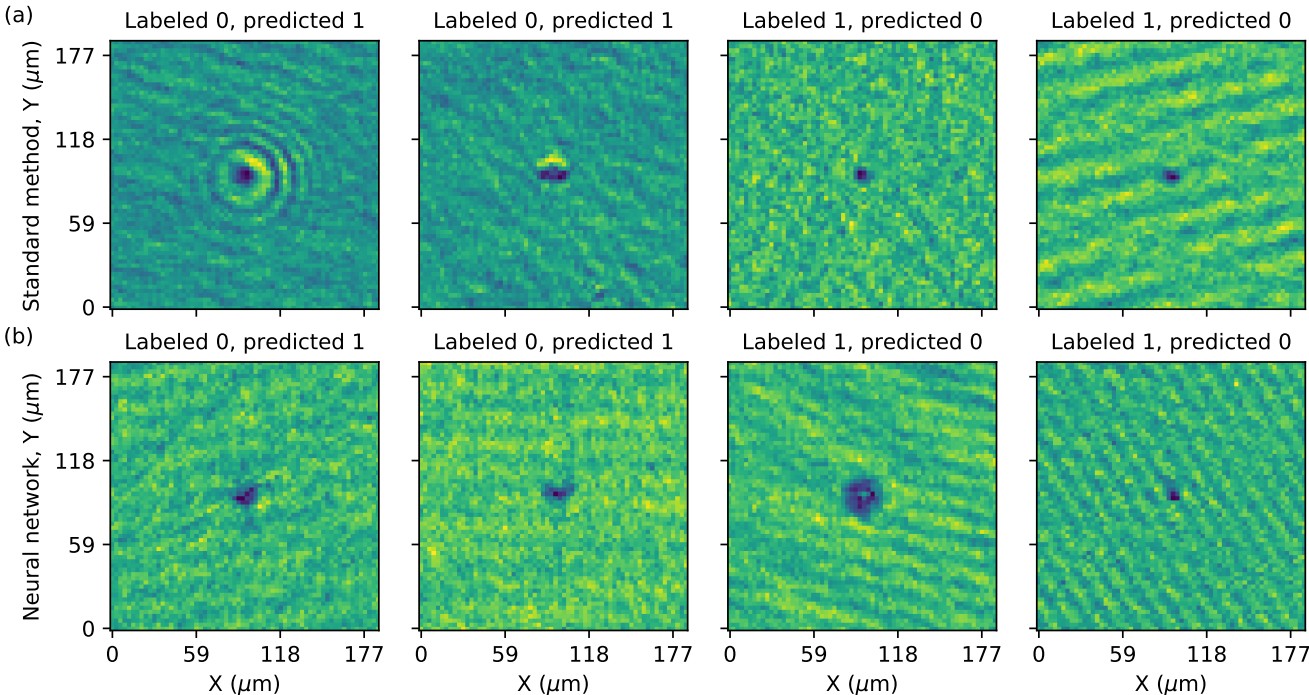

**Figure 13.** Examples of incorrect (false positive) or missed predictions (false negative) made by (a) the standard method and (b) the neural network model $N_{SH} = 1000$. The title in each panel shows the manual label and that determined by the model. False negatives indicated by 'labeled 0' while false positives are indicated by 'predicted 1'.

negative examples produced by the standard method were often particles with smaller diameters surrounded by distortion, as the examples in the figure illustrate.

The false positives produced by the $N_{SH} = 1,000$ model, such as the examples shown in Figures 10(b), were often close to a hologram edge, appeared blurry, and sometimes resembled a crescent moon in shape, rather than a well defined circle. Examples of false negatives in the figure show a particle appearing blurry with a less uniform (and dark) center, and a small particle surrounded by distortion. In some cases, the confidence by the reviewers was lower than 3 on the false predictions,
especially for those near the edge. Since manual labeling is also imperfect, it is possible that some of the more ambiguous images evaluated, in reality, contained a particle but that was slightly out-of-focus.

## 4 Discussion

In general, the observed differences between the $N_{SH} = 1,000$ model and the standard method on the HOLODEC test holograms demonstrates the advantages of using the convolutional-based neural networks in several key areas. First, it was much
more successful at differentiating reflections and other artefacts from in-focus particles. Second, the $N_{SH} = 1,000$ model was



more robust against making false negative predictions as caused by other kinds of distortion in the images, such as the patterns seen in the examples in Figure 13. This is reasonable as the neural model was optimized using both synthetic and HOLODEC examples, and with noise added to the synthetic images. Third, as noted, the neural network identified greater numbers of true particles from the HOLODEC holograms compared with the standard method. Fourth, the algorithm was designed so that

different components could be run in parallel. This parallelization is scalable so processing times can range from seconds to a few minutes per hologram depending on available resources. These results taken together demonstrate that the neural networks investigated here are capable of learning key features of images containing in-focus, localized particles, in both synthetic and real-world examples, and leveraging those parameterizations to make higher performing predictions on unseen holograms. Furthermore, the method we undertook here maintained a level of independence from the standard method processing package

which is beneficial for the evaluation of both methods. In this case it allowed us to identify issues with low density holograms in the current processing package.

However, more investigations with different sampling strategies and with higher density holograms are needed to determine this performance dependence and there may be modifications to the current archetecture that would reduce the issue of over prediction.

Another primary drawback with the supervised-learning approach pursued here was the non-existence of correctly labeled, real-world holograms obtained from HOLODEC. Using data generated from the physical model of the HOLODEC instrument proved to be insufficient as the sole source needed to produce a model that performed on the HOLODEC examples. We learned that adding noise to the synthetic images helped to improve performance, initially through trial and error. This was clearly not ideal and thus motivated perform manual labeling on HOLODEC images to optimize the noise transformations. Three problems

arise as a result of the lack of labeled HOLODEC images. The first is that we simply guessed which kinds of transformations to perform on the synthetic images. Second, the noise transformations reduced the model's ability to find the smallest diameter particles. Third, the manual evaluation for many examples with low confidence were ambiguous, and the label associated with each example did not represent the truth, but instead represented a mental model of what each reviewer thought an in-focus particle should look like in a 2D image plane. Furthermore, the total number of particles in the HOLODEC holograms was

not known in reality, only the examples that were identified by either method. Only physical measurements could be used to determine the true numbers. We should also note that manual training process may need to be repeated depending on the stability of the instrument's noise characteristics. It may be reasonable to expect the transformation optimization process would need to be conducted on each field project. Even the simplistic manual labeling we employed here is very labor intensive and is not realistic for routine deployments. For this reason, we are investigating less labor intensive methods for creating realistic

synthetic holograms.

What can be done in future work? First, a one-step approach for predicting particle coordinates and their shapes using only the reference hologram would provide the greatest benefit in terms of processing speed. Secondly, with the current approach involving wave-propagation, an object detector rather than the segmentation architecture may provide some benefit for obtaining $d$ for particles in higher density regions, because the bounding boxes can be overlapping and the area can be related to

the particle surface area. Thirdly, lowering the false alarm rate would help to solve problems that resulted from the match-





ing procedure. We only explored training models with one quarter of the examples being those that were thought would help with the false-positive issue, but informed up-sampling strategies should be considered in future investigations, such as aiming to optimize the ratio of positive to negative examples exposed to the model during training using ECHO rather than simply choosing that ratio to be one half. Another approach could train models using input sequences rather than a single image. For

example, common grid tiles at $z_{i-1}$, $z_i$, and $z_{i+1}$ could be combined into a single tensor with color dimension size 3 (rather than 1 as was used here) to make a mask prediction at $z_i$. Note also that recurrent-CNN methods are available, and the image transformer has recently been gaining popularity and can be used for video processing (Yan et al., 2021). The approach utilizes attention features for the space and time dimensions, that may be useful for identifying the correct $z$ for an in-focus particle. Such an approach could be applied to holograms to create 'videos' along the z-coordinate, using the 2D images created via

wave-propagation.

Lastly, potential ways to improve the models of the instruments could extend to using neural networks. The physics is not in question, rather how can we improve the characterization of uncertainties in the detector, such as potential optical imperfections, and other ways noise and impurities get represented in the instrument data. Guessing about the noise produced reasonable results when combined with extensive hyperparameter optimization and manual labeling efforts. However, a more

generic approach could be used to learn a more general model of instrument imperfections, that could be combined with the physical model to produce a more accurate representation of the HOLODEC instrument. For example, generative models such as adversarial networks and variational auto-encoders could be explored for obtaining a parameterized representation of real-world noise, that could be used to enhance the synthetic holograms to make them look more like the observed instrument outputs, but crucially, that can be prepared using exact positions and diameters for particles for training neural models.

## 5 Conclusions

In summary, this work describes a neural network hologram processing that provides a new approach for fast and accurate prediction of the particle locations and sizes in both simulated and real holograms obtained by HOLODEC. We should note that the full solution developed here does not represent an operational solution to HOLODEC processing, but it does layout a framework for such a solution. Components of this framework require further development. Simulated holograms were

produced using the physical model of the instrument for use as a truth data set to train models. However models trained only on the simulated data performed poorly on the holograms obtained by HOLODEC. The introduction of several types of pre-processing transformations, that included several types of noise added to training examples, enabled the discovery of model parameterizations that performed well on both simulated and real-world holograms. Two sets of HOLODEC holograms were labeled by NCAR scientists that were used to optimize the noise transformations and to compare the neural network model

against the standard method. Overall, the neural network outperformed the standard method at the task of particle identification by approximately 20% and predicted the particle location and shapes with high fidelity. The framework is also much faster compared to the standard method as it was designed to use GPUs, and so that analyzing wave-reconstructed planes at different $z$ values can be performed in parallel. Additionally, the Fourier transforms used during wave-propagation calculations utilized





GPU computation. Furthermore, extensive hyperparameter search was a crucial step in finding the best model. The introduction
of noise does not depend on the hologram data sets used here and could be applied to other data types where the physical model
of the instrument only represents ideal operation.

## A1 Average reconstruction and hologram processing time

In order to help accelerate processing time, we first considered how to speed up the wave propagation operation. An analysis
of different FFT packages presented on github (https://thomasaarholt.github.io/fftspeedtest/fftspeedtest.html) suggests that the
PyTorch FFT implementation is the fastest of those packages evaluated (numpy, TensorFlow, CuPy, PyFFTW). We ran a test
reconstructing 1000 HOLODEC planes and found on average the GPU implementation took 71 ms per plane while the CPU
implementation took 780 ms per plane. Thus, the PyTorch GPU implementation of the propagation step likely represents an
important speed improvement in processing HOLODEC data.

The neural network model ($N_{SH}$ = 1,000) was evaluated on NCAR's Casper supercomputer, using one core on a 2.3-GHz
Intel Xeon Gold 6140 processor (CPU) with 128 gigabytes of memory allocated, and an NVIDIA Tesla V100 32GB graphics
card (GPU). The standard method simulations were performed on NCARs Cheyenne supercomputer, on a node that contained
a 2.3-GHz Intel Xeon E5-2697V4 (Broadwell) processor (CPU). For the standard method, it took 3.5 core hours per hologram
on an ice case, while liquid cases varied by concentration and particle size, and ranged from 1-3 core hours per hologram. With
the GPU present, the neural network took approximately 2.1 CPU core hours per hologram, or about 7.5 seconds per plane. The
batch size used in the evaluation was set to 128 so that the total operation of HolodecML utilized nearly the maximum amount
of available GPU memory. Once all planes have been evaluated, the final 3D clustering step takes less than 0.1 seconds using
a threshold of $1000\,\mu m$. When sufficient computational resources are available, the parallel design and utilization of GPUs by
HolodecML enables processing speeds of less than 8 seconds per hologram. Serial inference time could be reduced further by
reducing the overlap between input tiles and by using smaller segmentation models.

## A2 Training and optimization performance

Table A1 lists the best parameters found for the $N_S$ = 1,000 and $N_{SH}$ = 1,000 models. The segmentation models considered
were the U-Net (Ronneberger et al., 2015a), U-Net++ (Zhou et al., 2018), MANet (Fan et al., 2020), LinkNet (Chaurasia and
Culurciello, 2017), FPN (Kirillov et al., 2017), PSPNet (Zhao et al., 2017), PAN (Li et al., 2018), Deeplabv3 (Chen et al.,
2017a) and Deeplabv3+ (Chen et al., 2018), while the encoder models considered were ResNet-18 and ResNet-152 (He et al.,
2016a), DenseNet-121 (Huang et al., 2017), Xception (Chollet, 2017), EfficientNet-b0 (Tan and Le, 2019), MobileNet ver-
sion 2 (Howard et al., 2017), DPN-68 (Chen et al., 2017b), and VGG-11 (Simonyan and Zisserman, 2014). See the package
segmentation-models-pytorch located at https://github.com/qubvel/segmentation_models.pytorch for more details on the seg-
mentation and encoder models. The training losses considered were the Dice loss, Dice combined with binary cross entropy
(BCE), intersection over union (IOU), Focal (Lin et al., 2017), Tyversky (Salehi et al., 2017), Focal-Tyversky, and the Lovasz-
Hinge loss (Berman et al., 2018). For additional definitions of each loss function, see the Holodec-ML software package





| Parameter | $N_S = 1{,}000$ | $N_{SH} = 1{,}000$ |
|---|---|---|
| Learning rate | $3.86 \times 10^{-4}$ | $2.46 \times 10^{-4}$ |
| Training loss | Focal-Tyversky | Focal-Tyversky |
| Segmentation model | U-Net | LinkNet |
| Encoder model | EfficientNet-b0 | Xception |
| Hologram transform | None | None |
| Tile transform | None | Normalized |
| Gaussian blur $\sigma$ | - | 2.125 |
| Gaussian noise | - | 0.326 |
| Brightness factor | - | 1.270 |

**Table A1.** The values of the best hyperparameters in the optimization studies for the neural segmentation models for the three species. The batch size was fixed at 16.

located at https://github.com/NCAR/holodec-ml. We always observed models that utilized pre-trained weights obtained from the ImageNet data set outperforming those that did not as well as requiring fewer training epochs.

Figure A1(a) plots the average dice coefficient computed on the validation set of synthetic images while (b) shows the same quantity computed with the binary-labeled HOLODEC examples, both versus epochs.

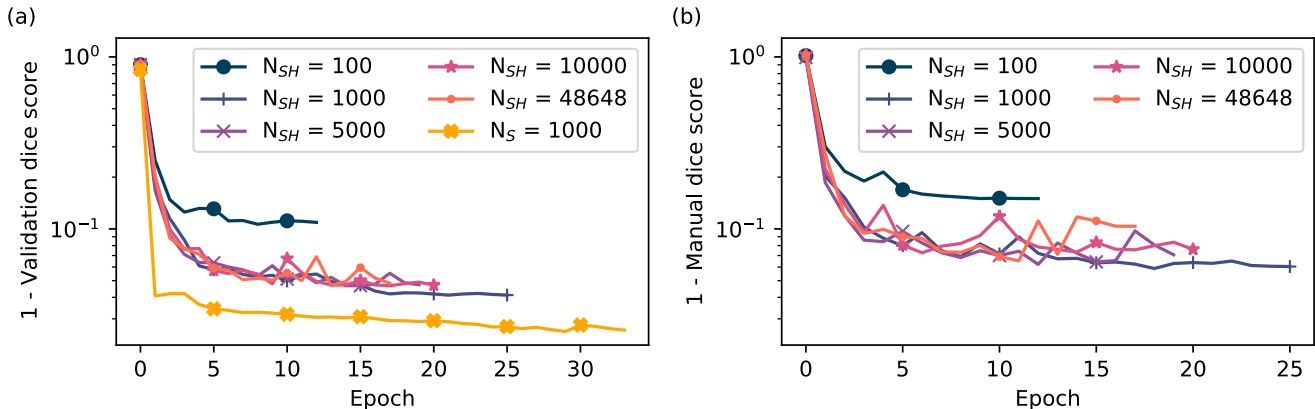

**Figure A1.** The average dice coefficient versus epochs computed on the validation set of training tiles and masks in (a), and on the validation set of human-evaluated HOLODEC images and binary labels in (b). Models trained and optimized with both the synthetic and the HOLODEC images are shown in (b).





|  | $N_S = 1{,}000$ | $N_{SH} = 100$ | $N_{SH} = 1{,}000$ | $N_{SH} = 5{,}000$ | $N_{SH} = 10{,}000$ | $N_{SH} = 48{,}648$ |
|---|---|---|---|---|---|---|
| x | 6.17 (82.30) | 24.36 (140.58) | 3.10 (40.18) | 5.89 (59.51) | 3.18 (31.10) | 3.11 (32.77) |
| y | 4.58 (55.66) | 23.63 (138.46) | 2.70 (22.64) | 5.21 (59.75) | 3.08 (30.32) | 3.07 (24.70) |
| z | 76.50 (123.17) | 83.52 (490.16) | 50.11 (63.07) | 45.52 (191.96) | 40.30 (142.44) | 50.42 (76.19) |
| d | 0.42 (1.16) | 1.47 (2.42) | 0.52 (1.34) | 1.57 (1.59) | 0.59 (1.56) | 0.69 (1.44) |
| RMSE | 91.44 (361.80) | 162.13 (756.47) | 59.14 (147.62) | 60.63 (313.17) | 48.22 (190.57) | 58.15 (141.11) |

**Table A2.** The mean and standard deviation in the absolute error for each $(x, y, z, d)$, and RMSE. The metrics were computed using the test set of synthetic holograms. All reported values have units of μm.

## A3  Additional results

Table A2 lists mean absolute error in each coordinate or particle diameter, as well as that for the computed RMSE. The standard deviation is listed in parenthesis. Figure A2 shows histograms for each coordinate and diameter for the false positive (unpaired) particles, computed using the test set of HOLODEC holograms. Table A3 lists the mean and standard deviation of the absolute error for each coordinate, diameter, and RMSE, computed using the manually evaluated test set of HOLODEC holograms.

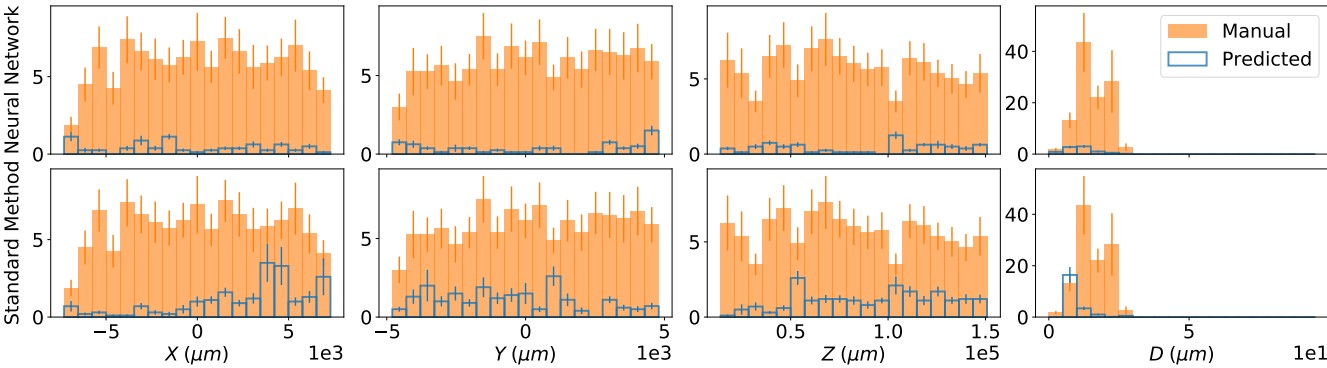

**Figure A2.** Histograms for $x$, $y$, $z$, and $d$ for false predictions are shown in each column, for the 10 testing HOLODEC holograms. The results for the standard method and the $N_{SH} = 1000$ model are shown in (a) and (b), respectively, relative to the true histograms. The value of each bin and the error bar were computed by taking the mean and standard deviation across the 10 test HOLODEC holograms.

*Acknowledgements.* This material is based upon work supported by the National Center for Atmospheric Research, which is a major facility sponsored by the National Science Foundation under Cooperative Agreement No. 1852977. We would like to acknowledge high-performance computing support from Cheyenne and Casper Computational and Information Systems Laboratory, CISL (2020) provided by NCAR's Computational and Information Systems Laboratory, sponsored by the National Science Foundation. The neural networks described here





|  | Standard | $N_{SH} = 1{,}000$ |
|---|---|---|
| x | 0.47, 3.84 | 0.44, 3.62 |
| y | 0.58, 4.40 | 0.50, 4.13 |
| z | 38.44, 37.65 | 33.64, 37.42 |
| d | 0.25, 0.45 | 0.28, 0.45 |
| RMSE | 45.06, 141.40 | 39.47, 133.15 |

**Table A3.** The mean and standard deviation in the absolute error for each $(x, y, z, d)$, and RMSE, for evaluation on the test set of HOLODEC holograms. All reported values have units of μm.

and simulation code used to train and test the models are archived at https://github.com/NCAR/holodec-ml. All HOLODEC and synthetic
hologram data sets created for this study are available at https://doi.org/10.5281/zenodo.6347222



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
