# Peer review of "Neural network processing of holographic images"

_Atmospheric Measurement Techniques, 2022_

## Referee Comment (RC1)

Review for "Neural network processing of holographic images" by Schreck et. al.

The manuscript "Neural network processing of holographic images" describes and evaluates a novel approach to processing holographic images using Machine Learning and Neural Networks. The authors used an existing dataset obtained with the holographic detector for clouds (HOLODEC) to compare current, standard methods to their new approach, and used synthetic holograms to train the neural network. The new approach has the potential to substantially speed up the data processing not only for HOLODEC, as shown here, but also other holographic imagers and therefore is useful for anyone applying holographic methods to in-situ or laboratory measurements. Although it is still not a fully automated process yet, this approach is very important and after further development may play a substantial role for future holographic data analysis. In general, the manuscript is well written, and I recommend publication with minor revisions.

*Main comments*

Line 180: You used 500 particles per hologram. However, holograms frequently have higher numbers of particles (few thousands). Why did you choose this number, and would the results differ if there were a factor of 10 more in each hologram?

You state that the synthetic holograms you used were corrupted by noise processes. It is unclear to me how exactly this was achieved. Why did you not use real empty holograms to inform what it would look like and use it as background for the synthetic holograms? That way you'd have the same background from the synthetic holograms as are present in reality. Since you explain towards the end of the manuscript you need to retrain the models before each field project anyways, you could include that way the field project specific noise.

Line 27/28: Where do these numbers come from? HOLODEC has a sample volume of ideally nearly 19 cm^3 (maximum x*y*z distance). Not all can be used, since particles are not uniformly detected, especially close to the edges. So, typically a sample volume of around 13 cm^3 has been used in the past, or even less, depending on the conditions. The sample area in x*y also decreases with distance z, leading to a cone shaped sample volume. This should be considered and explained. Similarly, each pixel is about 3 um wide, leading to the minimum detectable particle of about 6 um – but the resolution would still be 3 um.

*Suggestions for Figures for easier understanding*

Generally, Figures are good quality. However, some could be improved to guide the reader. Here are my suggestions:

None of the images have a colorbar, and while it may be acceptable in some cases, in examples such as e.g. Fig. 13 where particles are shown, it would be interesting to see if all panels have the same values or if the images were scaled.

Fig. 6: both panels in 6(a) have y-axes that have a gap between zero and the end. It makes it hard to see if the values in the plot are zero, or above. Either zero the axes or add gridlines for clarity.

Fig. 7/9: What exactly is the difference between Fig. 7 bottom row and Fig. 9 second row? The plots look identical to me and the description does not make it clearer. If they are the same, I'd suggest combining Figure 7 and 9 – if not, make clearer what the difference exactly is.

Fig. 11: (a) and (b) are mentioned in the caption, but not included in the figure.

Fig. 12:

- The colorbar is missing for all panels. The question is, is it the same colorbar in all figures or not? If it is, then it is noteworthy that the different approaches also size particles differently. In e.g. (c)(ii) the same particles than in (a) have a different color, and more orange/red particles are shown. Are the different methods influencing sizing that much?
- (a)(i) and (a)(ii) are the same – why not have it only once, maybe centered between the two rows? It took a while for me to realize they are the same and I looked for the differences.
- I find the naming (b)(i) and (b)(ii) in this case unnecessary and confusing, and naming (a)-(e) would work just as well here. But this is just a suggestion.
- It might be worth mentioning that the three axes are not equal size and therefore distances between particles are not to scale.

*Language*

Overall, the language is good. However, please check spelling and grammar carefully. Here is a list of things I found:

Line 220: "…number **of** particles…"

Line 241: "…number **of** pixels…"

Line 354: "…explicitly,_a transformation…" (space missing)

Line 412: "…trials…"

Line 465: $N_S$ is explained here, but $N_{SH}$ not. I could not find any specific definition for it in the manuscript.

Line 487: All abbreviations are explained here, but F1 score is not. I could not find any specific definition for it in the manuscript.

Line 643: "…architecture…"

---

## Author Response (AR1)

Reviews of amt-2022-97: Neural network processing of holographic images

**Response to Reviewer #1**

**Main comments**

**R1.1**: Line 180: You used 500 particles per hologram. However, holograms frequently have higher numbers of particles (few thousands). Why did you choose this number, and would the results differ if there were a factor of 10 more in each hologram?

**A1.1**: We originally started the project working with  $512 \times 512$  images that contained varying numbers of particles; the 500 came from scaling up the smaller images to the full-sized examples. We also tried to keep the computational time for the synthetic holograms reasonable. More particles can certainly be added in future investigations, especially if they are kept on the small size with a relatively narrow drop size distribution.

Regarding holograms with 10 times as many particles, we have added the following discussion to the Discussion section:

The performance of the  $N_{SH} = 1,000$  model on holograms containing thousands of particles, which was trained on holograms containing 500 particles, will depend on how the particles are distributed in 3D. The model should still predict masks around the clearly in-focus particles. As a result of matching/clustering procedure, however, these particles may be grouped into the same cluster and considered one (larger) particle with the clustering threshold of 1,000 um (this was observed even with holograms that contained hundreds of particles). It is possible that the model may still estimate the total mass but not the correct number. On the other hand, if the particles are sparsely distributed across (x,y,z) and on the smaller side, the model performance should be less dependent on the number of particles present.

**R1.2**: You state that the synthetic holograms you used were corrupted by noise processes. It is unclear to me how exactly this was achieved. Why did you not use real empty holograms to inform what it would look like and use it as background for the synthetic holograms? That way you'd have the same background from the synthetic holograms as are present in reality. Since you explain towards the end of the manuscript you need to retrain the models before each field project anyways, you could include that way the field project specific noise.

**A1.2**: We have added a clearer section number/heading, "2.5.1 Hologram image transformations", where the noise transformations are described. Several references to this section have been added in different areas in the revised manuscript to remind the reader what the transformations were and how they were used.

The background is an interesting concept. We had discussed this early on in the project and initially dismissed it because clearly some artifacts are not strictly constrained to intensity. For example, beam deviations from Gaussian and interference structure (due to clipping or

secondary reflections on the transmitter side) will result in both amplitude and phase fluctuations in the electric field interacting with the cloud particles. Due to the incomplete capture of the physics, we pursued a different approach to approximating non-ideal instrument behavior. However, we are actively working toward generating more realistic "noise" in our synthetic holograms, and the approach proposed here seems like a very reasonable approach to consider. Our view is that implementing this approach is out of scope with the work summarized in this manuscript, but we are very interested in leveraging this approach in our ongoing efforts.

**R1.3**: Line 27/28: Where do these numbers come from? HOLODEC has a sample volume of ideally nearly 19 cm3 (maximum  $x^*y^*z$  distance). Not all can be used, since particles are not uniformly detected, especially close to the edges. So, typically a sample volume of around 13 cm3 has been used in the past, or even less, depending on the conditions. The sample area in  $x^*y$  also decreases with distance z, leading to a cone shaped sample volume. This should be considered and explained. Similarly, each pixel is about 3 um wide, leading to the minimum detectable particle of about 6 um – but the resolution would still be 3 um.

**A1.3**: The quantities in question come directly from the referenced Spuler and Fugal Applied Optics (2011) publication. In the abstract it states:

"Experimental results demonstrate that the system is capable of recording holograms that can be reconstructed with resolution of better than 6.5 μm within a 15cm3 sample volume."

There should be some distinction made between the pixel resolution of the camera (3 um) and the optical system resolution (6.5 um) — AKA minimum resolvable spot size. This section of the text was referring to the latter. In order to clarify this, we added text to state the effective CCD pixel size is approximately 3 um.

**Minor comments**

**R1.4**: None of the images have a colorbar, and while it may be acceptable in some cases, in examples such as e.g. Fig. 13 where particles are shown, it would be interesting to see if all panels have the same values or if the images were scaled.

**A1.4**: In all figures showing holograms, we have changed the color scheme to grayscale to reflect the pixel scale 0-255 used throughout (for both HOLODEC and synthetic images), and added colorbars when appropriate.

**R1.5**: Fig. 6: both panels in 6(a) have y-axes that have a gap between zero and the end. It makes it hard to see if the values in the plot are zero, or above. Either zero the axes or add gridlines for clarity.

**A1.5**: We have added a grid-line at 0 particles and updated the caption in Figure 6 to point out the two grid-lines correspond with 0 and 500 particles.

**R1.6**: Fig. 7/9: What exactly is the difference between Fig. 7 bottom row and Fig. 9 second row? The plots look identical to me and the description does not make it clearer. If they are the same,

I'd suggest combining Figure 7 and 9 – if not, make clearer what the difference exactly is.

**A1.6:** They are the same. We have left Figures 7 and 9 as they are but added a sentence in section 3.4 to clarify that the panels are the same:

"For clarification, the N\_SH = 1000 results in Figure 7 (bottom row) are shown again in Figure 9 (second row)."

**R1.7**: Fig. 11: (a) and (b) are mentioned in the caption, but not included in the figure.

A1.7: (a) and (b) have been added to both Fig. 11 and Fig. A2

**R1.8: Fig 12:**

- The colorbar is missing for all panels. The question is, is it the same colorbar in all figures or not? If it is, then it is noteworthy that the different approaches also size particles differently. In e.g. (c)(ii) the same particles than in (a) have a different color, and more orange/red particles are shown. Are the different methods influencing sizing that much?

- (a)(i) and (a)(ii) are the same – why not have it only once, maybe centered between the two rows? It took a while for me to realize they are the same and I looked for the differences.

- I find the naming (b)(i) and (b)(ii) in this case unnecessary and confusing, and naming (a)-(e) would work just as well here. But this is just a suggestion.

- It might be worth mentioning that the three axes are not equal size and therefore distances between particles are not to scale.

**A1.8:** We have added a color bar, which applies to all panels in Figure 12. The coloring convention is also consistent across all panels. We also have made all particle symbol sizes the same so that the color bar alone instructs the reader on the size of the particles.

We have removed the duplicated "True" particles 3D plot, and relabeled the five panels (a) through (e). Finally, we added a line to the caption in Figure 12 stating: "Note that the three axes do not have the same size, therefore distances between particles do not scale."

R1.9: Line 220: "...number of particles..."

A1.9: Corrected.

R1.10: Line 241: "...number of pixels..."

A1.10: Corrected.

**R1.11**: Line 354: "...explicitly,\_a transformation..." (space missing)

A1.11: Corrected.

R1.12: Line 412: "...trials..."

**A1.12: Corrected.**

**R1.13**: Line 465: N\_S is explained here, but N\_SH not. I could not find any specific definition for it in the manuscript.

**A1.13**: We have rewritten the section including line 465 to clarify the difference between "S" and "SH" notation used in the results presentation:

"We investigated six models to probe the performance dependence on the choice of  $N\$  as well as the noise introduced during training as described in Section~\ref{sec:image\_trans}. We use a subscript `S' to reference the first model that was trained and optimized on synthetic holograms only, as described in Section~\ref{sec:opt}, using  $N_S$  = 1,000 bins along the z direction. The remaining models used different resolutions along z, which were  $N_{SH}$  = 100, 1,000, 5,000, 10,000, and 46,648, where `SH' means the models were trained on synthetic holograms that were corrupted by noise processes. We optimized the corruption in hyperparameter optimization by utilizing the manually labeled validation HOLODEC examples, hence the subscript `SH', which is used to differentiate these models from the `S' model that is used here as a baseline."

**R1.14**: Line 487: All abbreviations are explained here, but F1 score is not. I could not find any specific definition for it in the manuscript.

**A1.14**: At the end of section 2.4.4 we have updated the following sentence: "The paired particles can be used to compute performance metrics such as accuracy and F1 score (defined here as the harmonic mean between precision and recall), while the predicted particle numbers allow us to construct a contingency table for the holograms."

R1.15: Line 643: "...architecture..."

A1.15: Corrected.

**Response to Reviewer #2**

The manuscript presents a novel approach for processing data from holographic imagers called HolodecML. HolodecML utilizes GPU hardware to accelerate the reconstruction of the holograms. A neuronal network is trained on patches of the reconstructed images to detect the position of particles. For training, synthetic holograms are created and corrupted to mimic noise that appears in holograms taken by the actual HOLODEC instrument.

The presented utilization of a neuronal network for processing holograms is a very innovative approach, which has great potential to improve the data analysis of holographic images. Additionally, the generation of synthetic holograms with realistic noise is a novelty.

**R2.1:** As these synthetic holograms can be used as ground truth for validating processing approaches because the position and size of the particles are known, the potential of the synthetic holograms could be discussed more prominently.

**A2.1**: We appreciate that you recognize this fact. We mention this in a few places, but it seems reasonable to further emphasize it. To be clear, there are trade-offs in using simulations for this purpose. In particular, overly idealized instrument models are generally not highly performant on real data. In section 2.2 we added the following text to attempt to emphasize this:

"As previously noted, the major benefit to using synthetic holograms for training is that particle positions and size are known and therefore the machine learning solution will not inherit errors or biases from standard processing of actual data. However, the challenge in using synthetic data is that simulations generally fail to fully capture non-ideal aspects of instrument operation, which can impact the effectiveness of the machine learning solution when it is deployed to actual data."

**R2.2:** In general, the manuscript is well written and the approach is explained in an understandable way. Acknowledging the fact that the authors have the difficult task of presenting a complicated issue at the intersection of computer science and atmospheric science, it still took some time to understand the main message of the manuscript.

In my eyes, the manuscript could be strengthened by focusing on what are the practical advantages of HolodecML in the analysis of holographic data and what is the potential of the new approach. The part of the analysis that is not central to the main points (e.g. analysis with the number of z-slides not equal 1000) should be moved to the Appendix, whereas the performance comparison should be moved to the main manuscript. The standard approach should be described in more detail to be able to understand the changes and results of the HolodecML approach. For clear reference, the training and test datasets could be given a unique name and their properties could be summarized in a table.

**A2.2**: We feel it is important to establish that the machine learning solution actually works, which is why we have emphasized these metrics and methods in the body of the manuscript. Furthermore, we felt it was important to be clear about the various operational and design tradeoffs in the solution (e.g. the number of z-slides not equal to 1000) because these offer potential performance trades (e.g. speed vs accuracy). This is in contrast to what is often employed in machine learning publications, where the final solution is provided but solution space is rarely described and therefore, not useful to anyone who is not trying to solve the exact same problem.

We uploaded the datasets and a description of them to zenodo. A DOI (https://doi.org/10.5281/zenodo.6347222) linking to the datasets is listed in the Acknowledgements section. Additionally, we added Appendix A2 "Data sets" and Table A2 which links the named data files available from the DOI with the named training data splits used in the paper.

Main comments:

**R2.3:** The HolodecML approach requires the same reconstruction of planes through wave propagation as the standard approach. Could the GPU implementation of the reconstruction introduced by HolodecML also be implemented in the standard approach? If yes, the performance gains would be of great benefit to the community using the standard approach. Therefore, the performance gains during reconstruction should be separately discussed from the computational cost of the neuronal network for particle detection and the discussion should be moved from the Appendix to the main manuscript.

**A2.3**: Yes, HoloSuite could also leverage this speed up. To clarify this, we moved the content on the GPU acceleration of wave propagation to the section "Estimating \$z\$ through wave propagation" and added a comment that GPU acceleration could also be applied to HoloSuite. We then moved the discussion on processing speed to the results section.

**R2.4**: I have my doubt that the statement that HolodecML improves particle detection by 20% is based on a fair comparison. The particle detection in the standard approach is normally tuned rather loose to ensure that all particles are detected (i.e., minimizing the number of false negatives). Consequently, a higher number of artifacts that are not real particles (i.e., a higher number of false positives) are detected. The reason behind this is, that particles that are not detected cannot be retrieved at a later stage, but artifacts can be sorted out by a classification algorithm (e.g., by a neuronal network as described in Touloupas et. al, 2020). Therefore, the large false-positive rate and low false negative rate presented in Figure 10 is expected in the standard approach and could be improved by a classification algorithm. How was the particle detection optimized in the standard approach? Were artifacts sorted out by a classification algorithm?

**A2.4**: The reviewer brings up a reasonable point which we struggled with throughout the writing of this manuscript. HoloSuite is modular and has a number of settings that can produce very different results. As a result, it is very difficult to make a complete comparison to the package. To be clear, the intent of this manuscript is not to declare HolodecML as "better" than HoloSuite, but rather highlight potential approaches that (might very well as likely) be integrated into any processing approach. The comparison presented here represents data from the released CSET dataset. Because this dataset is from a quality controlled field project, we feel that it is a reasonable basis for comparison. We certainly acknowledge that this does not make it the best possible results from the processor, but it does represent a practical result.

We should also highlight the fact that the HoloSuite results also show a higher false negative rate than HolodecML. While this error is smaller than false positives, and, unlike false positives, reduces with increased label confidence, it remains an important and notable metric of performance evaluation, suggesting that the difference between the two methods cannot be entirely made up with a second processing pass on the HoloSuite output.

We have further revised the manuscript to mention that false positives could be filtered out with a second stage classification algorithm in subsection "The standard method and \$N\_{SH}\$ performance on HOLODEC holograms": "We should note that false positives in the standard method could be improved by adding a second pass classifier to eliminate artifacts from the data. While the data processing considered here did not include this second pass, we still believe it represents a reasonable baseline for comparison since it is part of a released processing run."

Minor comments:

**R2.5**: Figure 1 (a): The interference fringes are almost invisible.

**A2.5**: We have updated all hologram images in the revised manuscript to use a gray-scale for the pixel values (which may range from 0-255 in all images). The HOLODEC example shown in Figure 1 should better show the interference fringes.

R2.6: Line 166: What is the RF07 subset?

**A2.6**: In section 2.2, we have clarified that "RF07 refers to 'Research Flight #7', which occurred on 19 July 2015 over the Pacific Ocean between Kona Hawaii and Sacramento CA." For more details on the CSET data set, the reader may refer to Albrecht et al., 2019.

R2.7: Line 167: Was HolodecML able to detect the few ice crystals?

**A2.7**: We did not see any while manually labeling the 20 examples from RF07, but we have other datasets where they are present that we are using to extend the modeling approach. We have added the following sentence to the Discussion to note that the current approach could be extended to detect, for example, ice crystals using a multi-categorical segmentation model:

"Finally, the binary prediction task we selected for the segmentation models could be extended to K label types so that other objects such as ice crystals could be identified and characterized"

R2.8: Line 201: Is the reference to Figure 3(c)(v) correct?

A2.8: We thank the reviewer for pointing out this mistake, the corrected reference is 3(c)(vi).

**R2.9**: Line 203-210: The creation of an independent processing approach is an important motivation for your work and should already be discussed in the introduction section.

**A2.9**: We have merged lines into the introduction, which is now the second-to-last paragraph:

"Additionally, it was decided that there would be significant benefit in developing HolodecML independently of the current state-of-the-art processing software, referred to here as the ``standard method" (and which is discussed below). The motivation for this is twofold. First, by creating an independent processing approach, HolodecML can help identify possible biases and sources of error in the current processing package. Second, this avoided creating a solution where the standard method imposed a ceiling on the processor performance."

The second paragraph in Section 2.4 now begins as:

"In order to develop a processor independent of the standard method, we had to develop a training approach, illustrated in Figure 3(b), that avoided excessive manual labeling (i.e. it is unrealistic to conduct manual labeling of particle position and size over large datasets and the accuracy of such approaches would likely be suspect)."

**R2.10**: Line 261: In the N = 48648 case the distance between planes is around 3 um. Why did you consider such a large number of planes if you expect limited performance improvement below the in-depth resolution of 57 mm?

**A2.10**: As the reviewer states, the impact of reconstructing more planes at finer scales than the DOF of the instrument is not expected to produce a significant benefit to the results. However, since we did do the analysis, it seemed worthwhile to include it in the results. Our hope is that this will benefit readers that are not entirely familiar with the concept of DOF. The results of the analysis confirm that adding more planes have diminishing returns for instrument performance.

**R2.11**: Figure 5: The thresholding in the standard method should detect the particle in all three planes. Why is the mask not visible in all three planes?

**A2.11**: We thank the reviewer for pointing this out. We have added the following sentence to the caption in Figure 5: "The standard method prediction is the result of a clustering procedure that eliminates the particle in multiple planes."